# A Mathematics-Inspired Learning-to-Optimize Framework for Decentralized Optimization

## Abstract

Most decentralized optimization algorithms are handcrafted. While endowed with strong theoretical guarantees, these algorithms generally target a broad class of problems, thereby not being adaptive or customized to specific problem features. This paper studies data-driven decentralized algorithms trained to exploit problem features to boost convergence. Existing learning-to-optimize methods typically suffer from poor generalization or prohibitively vast search spaces. In addition, the vast search space of communicating choices and final goal to reach the global solution via limited neighboring communication cast more challenges in decentralized settings. To resolve these challenges, this paper first derives the necessary conditions that successful decentralized algorithmic rules need to satisfy to achieve both optimality and consensus. Based on these conditions, we propose a novel **M**athematics-**i**nspired **L**earning-to-**o**ptimize framework for **D**ecentralized **o**ptimization (**MiLoDo**). Empirical results demonstrate that MiLoDo-trained algorithms outperform handcrafted algorithms and exhibit strong generalizations. Algorithms learned via MiLoDo in 100 iterations perform robustly when running 100,000 iterations during inferences. Moreover, MiLoDo-trained algorithms on synthetic datasets perform well on problems involving real data, higher dimensions, and different loss functions.

## 1 Introduction

With the ever-growing scale of data and model sizes in modern machine learning and optimization, there is an increasing demand for efficient distributed algorithms that can harness the power of multiple computing nodes. Traditional centralized approaches that rely on global communication and synchronization face significant communication overhead and latency bottlenecks. This challenge has given rise to decentralized learning, an emerging area that promises to alleviate these issues.

In decentralized learning, computing resources like CPUs/GPUs (known as nodes) are connected via a network topology and only communicate with their immediate neighbors, averaging model parameters locally. This neighbor-based averaging eliminates the need for global synchronization, drastically reducing communication costs compared to centralized methods. Moreover, decentralized algorithms exhibit inherent robustness, maintaining convergence despite node or link failures, as long as the network remains connected. Decentralized optimization has emerged as a standard paradigm for distributed training without centralizing data (Liu et al., 2024), offering significant advantages in communication efficiency (Lian et al., 2017) and privacy protection (Yu et al., 2024), making it a promising approach for privacy-preserving distributed learning across data centers.

**Motivations for data-driven decentralized algorithms.** Most existing decentralized algorithms are *handcrafted*, driven by optimization theories and expert knowledge. Notable examples include primal algorithms such as DGD (Nedic & Ozdaglar, 2009; Yuan et al., 2016) and Diffusion (Lopes & Sayed, 2008; Chen & Sayed, 2012), dual algorithms like dual averaging (Duchi et al., 2011), and primal-dual algorithms such as decentralized ADMM (Shi et al., 2014), EXTRA (Shi et al., 2015a), Exact-Diffusion (Yuan et al., 2018b) (also known as NIDS (Li et al., 2019)), and Gradient-Tracking (Nedic et al., 2017; Xu et al., 2015; Di Lorenzo & Scutari, 2016). These handcrafted decentralized algorithms are designed to address a wide range of optimization problems, making them versatile and broadly applicable. Furthermore, their convergence guarantees are valid in worst-case scenarios, ensuring strong reliability. However, due to their emphasis on theoretical guarantees and broad

applicability, handcrafted algorithms often fail to leverage problem-specific features in data and thus exhibit sub-optimal performance in practice. This motivates us to explore *data-driven* decentralized algorithms that exploit problem-specific features to improve performance.

**Learning to Optimize (L2O).** Our main idea draws inspiration from the L2O paradigm (Gregor & LeCun, 2010; Andrychowicz et al., 2016; Bengio et al., 2021; Monga et al., 2021; Chen et al., 2022) that utilizes machine learning techniques to develop optimization algorithms (also known as "*optimizers*"). Specifically, L2O employs a data-driven procedure where an optimizer is trained by its performance on a set of representative example problems (which we call "*optimizees*"). Through this training process, the learned optimizer becomes tailored and adaptive to the structures of problems similar to those in the training set, potentially outperforming general-purpose, handcrafted algorithms.

Two mainstreams in L2O are algorithm unrolling (Gregor & LeCun, 2010; Monga et al., 2021) and the generic L2O (Andrychowicz et al., 2016). Algorithm unrolling conceptualizes each iteration of a certain hand-crafted optimization algorithm as a layer in a neural network, inducing a feed-forward network. In contrast, the generic L2O does not impose any prior mathematical knowledge on the optimizer to be learned. Instead, it crudely parameterizes the optimizer with a recurrent neural network and learns it through end-to-end training.

**Challenges in applying L2O to decentralized optimization.** While algorithm unrolling and generic L2O have shown strong empirical successes (Andrychowicz et al., 2016; Lv et al., 2017; Wichrowska et al., 2017; Wu et al., 2018; Metz et al., 2019; Chen et al., 2020a; Micaelli & Storkey, 2021; Metz et al., 2022b; Liu et al., 2023; Gregor & LeCun, 2010; Moreau & Bruna, 2017; Chen et al., 2018; Liu & Chen, 2019; Ito et al., 2019; Yang et al., 2016; Zhang & Ghanem, 2018; Adler & Öktem, 2018; Solomon et al., 2019), their direct application to decentralized settings poses several challenges.

- **Memory bottleneck.** Algorithm unrolling requires storing a neural network with as many layers as optimization iterations, which easily exhausts available memory, especially as the number of unrolled iterations increases. This situation becomes even more challenging in decentralized optimization, where researchers are usually constrained to test their algorithms on simple target problems (Nedic et al., 2017; Shi et al., 2014; Xu et al., 2015; Shi et al., 2015a; Yuan et al., 2018b) within dozens of nodes (Chen & Sayed, 2012; Wang et al., 2021; Shi et al., 2015b; Xu et al., 2015; Nedic et al., 2017), and we shall maintain much more memory than traditional decentralized algorithms during the training stage.

- **Vast search space.** While more memory-efficient, the generic L2O faces a significant challenge: how to parameterize an optimizer properly. The parameter space of the generic L2O is vast, rendering its training highly ineffective. This challenge is exacerbated in decentralized optimization due to the extra need to learn inter-node interaction (*e.g.*, when and with whom to communicate, and what information to exchange), thereby further expanding the parameter space.

- **Consensus constraint.** In decentralized optimization, nodes must achieve consensus through local communication with immediate neighbors. This consensus constraint incurs significant complexity to L2O, as nodes, with trained optimizers, must adapt their behaviors to ensure convergence towards a *common* solution despite lacking global communication.

- **Weak generalization.** L2O often struggles to generalize to out-of-distribution tasks. Without theoretical guidance, it is challenging for L2O to handle the more sophisticated loss landscapes encountered in unseen problems. Refer to Section 4.4 and Figures 7 and 8 in Chen et al. (2022) This challenge naturally carries over in decentralized L2O.

**Contributions.** To address the aforementioned challenges, this paper proposes a novel **M**athematics-**i**nspired **L**earning-to-**o**ptimize framework for **D**ecentralized **o**ptimization (**MiLoDo**). MiLoDo adopts the generic L2O strategy to circumvent memory bottlenecks. However, instead of learning an optimizer directly from an unconstrained parameter space, we introduce mathematical structures inherent in decentralized optimization to guide MiLoDo's learning process. This significantly narrows the parameter space, enforces asymptotic consensus among nodes, and ensures generalization across out-of-domain tasks. Our contributions can be summarized as follows:

- We derive fundamental mathematical conditions that learning frameworks for decentralized optimization that converges quickly to the exact solution should satisfy. These conditions will serve as guiding principles for training decentralized optimizers that can achieve consensus and optimality.

- Building upon these conditions, we derive a math-inspired neural network structure for MiLoDo. Utilizing this structure, we demonstrate that for MiLoDo-trained optimizers, any fixed point attains consensus across nodes and achieves the solution to the target decentralized problem.
- We develop effective training strategies for MiLoDo, which are critical to ensuring the fast and robust convergence of learned optimizers. We conduct extensive experiments to validate the strong generalization and superior convergence of MiLoDo-trained optimizers.

**Experimental results.** Our experimental results demonstrate that MiLoDo-trained optimizers exhibit strong generalization to out-of-distribution tasks. Specifically, they can adapt to tasks with varying data distributions, problem types, and feature dimensions. For instance, in the high-dimensional LASSO problem illustrated in Fig. 1, while trained to operate for 100 iterations when solving prob-

lems in the training dataset, the MiLoDo-trained optimizer performs exceptionally well for way more iterations (*e.g.*, 200,000 iterations) when solving unseen problems in the test dataset. Furthermore, our observations indicate that even when trained on LASSO problems of dimension 300, these optimizers can proficiently solve the LASSO problem with dimension 30,000, as shown in Fig. 1. More impressively, MiLoDo-trained optimizer achieves about $1.5\times \sim 2\times$ speedup in convergence compared to state-of-the-art handcrafted decentralized algorithms. These phenomena justify the necessity to incorporate mathematical structures into L2O for decentralized optimization.

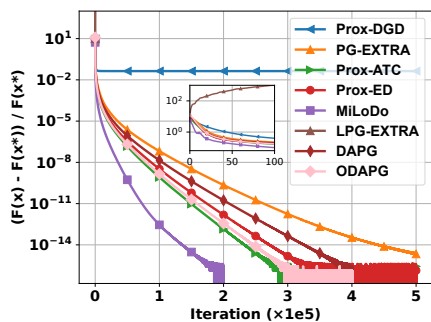

Figure 1: Numerical comparison between decentralized algorithms in solving LASSO problems of dimension 30,000. Experimental details are deferred to Sec. 5.

**Related work on decentralized L2O.** Previous studies have designed various L2O algorithms for decentralized optimization, most of which are based on algorithm unrolling. Kishida et al. (2020) and Ogawa & Ishii (2021) use algorithm unrolling to learn decentralized algorithms for the consensus problem. Noah & Shlezinger (2023) unrolls D-ADMM, while Wang et al. (2021) unrolls prox-DGD and PG-EXTRA. Hadou et al. (2023) proposed an unrolled algorithm called U-DGD. However, none of these learned optimizers can operate for more than 100 iterations due to the explosive memory cost caused by algorithm unrolling. Additionally, a recent work Zhu & Lu (2023) employs a reinforcement learning agent to control local update rules through a coordinator linked with all computing nodes, which is not fully decentralized. More related works on decentralized optimization and learning-to-optimize are discussed in Appendix A.

## 2 PRELIMINARIES

**Generic L2O.** Let's begin by addressing a fundamental inquiry: Given a set of optimization problems $\mathcal{F}$, how can we learn an optimizer from this dataset? Consider a parameterized optimizer seeking to minimize $\min_{\boldsymbol{x}} f(\boldsymbol{x})$, represented as:

$$\boldsymbol{x}^{k+1} = \boldsymbol{x}^k + \boldsymbol{m}^k(\nabla f(\boldsymbol{x}^k); \boldsymbol{\theta}), \quad k = 0, 1, \cdots, K-1 \tag{1}$$

where $\boldsymbol{m}^k$ is a learnable update rule typically implemented as a deep neural network[1] parameterized by $\boldsymbol{\theta}$. To determine $\boldsymbol{\theta}$, we evaluate and refine the performance of the optimizer (1) over the initial $K$ steps on the dataset $\mathcal{F}$. Specifically, this entails minimizing a loss function $\mathbb{E}_{f \in \mathcal{F}}[\frac{1}{K} \sum_{k=1}^{K} f(\boldsymbol{x}^k)]$. This loss minimization process is termed *training an optimizer*, with the employed dataset $\mathcal{F}$ referred to the training set. Upon determining $\boldsymbol{\theta}$, the learned update rule $\boldsymbol{m}^k(\cdot, \boldsymbol{\theta})$ will map the gradient $\nabla f(\boldsymbol{x}^k)$ to a desirable descent direction per iteration. Compared to standard gradient-based algorithms, such an optimizer is tailored to $\mathcal{F}$ and ideally exhibits faster convergence on unseen problems similar to those in the training set. In this context, the generalization of a trained optimizer involves two aspects: *generalizing to iterations beyond $K$* and *generalizing to diverse problems $f \notin \mathcal{F}$*.

**Decentralized optimization.** In this paper, we aim to learn a decentralized optimizer. To formally define decentralized optimization, we first introduce several definitions used throughout this paper.

---

[1]A common approach is using Recurrent Neural Network (RNN) $\boldsymbol{\phi}$: $(\boldsymbol{m}^k, \boldsymbol{h}^k) = \boldsymbol{\phi}(\nabla f(\boldsymbol{x}^k), \boldsymbol{h}^{k-1}; \boldsymbol{\varphi})$, where $\boldsymbol{h}^k$ is the hidden state at iteration $k$, and $\boldsymbol{\varphi}$ represents the learnable parameters in the RNN model $\boldsymbol{\phi}$.

**Definition 1** (Decentralized network topology)**:** We use $\mathcal{G} = (\mathcal{V}, \mathcal{E})$ to denote the undirected network topology in a decentralized system, where $\mathcal{V} = \{1, 2, \cdots, n\}$ represents the set of all $n$ nodes and $\mathcal{E} = \{\{i, j\} \mid i \neq j, \text{ node } i \text{ can communicate with node } j\}$ denotes the set of all edges.

Throughout this paper, we assume the decentralized network is strongly connected, meaning there is always a path connecting any pair of nodes in the network topology.

**Definition 2** (Families of objective functions)**:** We define the following function families:

$$\mathcal{F}(\mathbb{R}^d) = \left\{ f : \mathbb{R}^d \to \mathbb{R} \mid f \text{ is closed, proper and convex} \right\},$$

$$\mathcal{F}_L(\mathbb{R}^d) = \left\{ f : \mathbb{R}^d \to \mathbb{R} \mid f \text{ is convex, differentiable and } L\text{-smooth} \right\}.$$

This paper targets to solve the following problem over a network of $n$ collaborative computing nodes:

$$\boldsymbol{x}^\star = \arg \min_{\boldsymbol{x} \in \mathbb{R}^d} \left\{ \frac{1}{n} \sum_{i=1}^n f_i(\boldsymbol{x}) + r(\boldsymbol{x}) \right\} \tag{2}$$

Here, the local cost function $f_i(\boldsymbol{x}) \in \mathcal{F}_L(\mathbb{R}^d)$ is privately maintained by node $i$, and $r(\boldsymbol{x}) \in \mathcal{F}(\mathbb{R}^d)$ is a regularization term shared across the entire network. We assume each node $i$ can locally evaluate its own gradient $\nabla f_i(\boldsymbol{x})$ and must communicate to access information from other nodes. Additionally, communication is confined to the neighborhood defined by the underlying network topology; if two nodes are not direct neighbors, they cannot exchange messages.

**A naive approach to extend generic L2O to decentralized optimization.** A straightforward approach to extend generic L2O (1) to the decentralized setting is as follows:

$$\boldsymbol{x}_i^{k+1} = \boldsymbol{x}_i^k + \boldsymbol{m}_i^k \Big( \{\boldsymbol{x}_j^k, \nabla f_j(\boldsymbol{x}_j^k)\}_{j \in \mathcal{N}(i) \cup \{i\}}; \boldsymbol{\theta}_i \Big), \quad \forall i \in \mathcal{V}, \tag{3}$$

where $\boldsymbol{x}_i^k$ represents the local variable maintained by node $i$ at iteration $k$, $\boldsymbol{m}_i$ denotes a learnable update rule with parameter $\boldsymbol{\theta}_i$ retained by node $i$, and notation $\mathcal{N}(i)$ signifies the set of immediate neighbors of node $i$. While general, the naive update rule $\boldsymbol{m}_i$ encounters two significant challenges: **(I)** its vast search space, as finding an effective $\boldsymbol{m}_i$ requires exploring all possible combinations of the iterative variables, gradients, and neighbors' information. This complexity makes it difficult to identify a suitable rule, especially when training samples are limited. **(II)** The update rule in (3) lacks a mechanism to ensure that all nodes reach consensus and converge to the common solution of problem (2), i.e., $\boldsymbol{x}_1^\star = \cdots = \boldsymbol{x}_n^\star = \boldsymbol{x}^\star$, where $\boldsymbol{x}_i^\star$ represents the limit of sequence of $\{\boldsymbol{x}_i^k\}_{k=1}^\infty$. These two limitations inspire us to introduce mathematical structures into $\boldsymbol{g}_i$ to narrow the search space and enforce asymptotic consensus among nodes.

## 3 MATHEMATICS-INSPIRED UPDATE RULES FOR DECENTRALIZED OPTIMIZATION

This section establishes the mathematical principles underlying decentralized optimization and utilizes them to motivate the learning-to-optimize update rules for decentralized optimization. In the subsequent subsections, we will first determine the base update rules that decentralized optimizers should follow, and then specify the concrete structure for each base rule.

### 3.1 BASE UPDATE RULES

The base rule serves as a fundamental mechanism to update optimization variables. While it delineates the necessary inputs, it does not specify a particular structure that the rule must follow. Examples include $\boldsymbol{m}(\cdot, \boldsymbol{\theta})$ in generic L2O (1) and $\boldsymbol{m}_i(\cdot, \boldsymbol{\theta}_i)$ in naive decentralized L2O (3). This subsection aims to identify improved base update rules that resolve the aforementioned issues.

**Decentralized optimization interpreted as constrained optimization.** One limitation in the naive update rule in (3) is that it cannot explicitly enforce variable consensus during updates. To address this limitation, we reformulate the unconstrained problem (2) as the following constrained optimization:

$$\min_{\boldsymbol{X} \in \mathbb{R}^{n \times d}} \quad \frac{1}{n} \sum_{i=1}^n f_i(\boldsymbol{x}_i) + r(\boldsymbol{x}_i), \quad \text{s.t.} \quad \boldsymbol{x}_i = \boldsymbol{x}_j, \, \forall \{i, j\} \in \mathcal{E}. \tag{4}$$

Here, the optimization variable $\boldsymbol{X} = [\boldsymbol{x}_1^\top, \boldsymbol{x}_2^\top, \cdots, \boldsymbol{x}_n^\top]^\top \in \mathbb{R}^{n \times d}$ stacks the local variables across all nodes. The consensus constraints in (4) are imposed according to the structure of the underlying network topology. Since the network is strongly connected, we have $\boldsymbol{x}_1 = \boldsymbol{x}_2 = \cdots = \boldsymbol{x}_n$.

**Primal-dual algorithm and its implication.** The Lagrangian function of (4) is given as follows

$$\mathcal{L}(\boldsymbol{X}, \{\boldsymbol{\nu}_{i,j}\}_{\{i,j\}\in\mathcal{E}}) = \frac{1}{n}\sum_{i=1}^{n}(f_i(\boldsymbol{x}_i) + r(\boldsymbol{x}_i)) + \sum_{\{i,j\}\in\mathcal{E}}\langle\boldsymbol{\nu}_{i,j}, \boldsymbol{x}_i - \boldsymbol{x}_j\rangle, \tag{5}$$

with $\boldsymbol{\nu}_{i,j}$ the dual variable of constraint $\boldsymbol{x}_i = \boldsymbol{x}_j$. The primal-dual algorithm to solve (4) is given by:

$$\boldsymbol{x}_i^{k+1} = \boldsymbol{x}_i^k - \frac{\gamma}{n}\left(\nabla f_i(\boldsymbol{x}_i^k) + \boldsymbol{g}_i^{k+1} + \boldsymbol{y}_i^k\right), \quad \boldsymbol{g}_i^{k+1} \in \partial r(\boldsymbol{x}_i^{k+1}), \tag{6}$$

$$\boldsymbol{y}_i^{k+1} = \boldsymbol{y}_i^k + 2\gamma\sum_{j\in\mathcal{N}(i)}(\boldsymbol{x}_i^{k+1} - \boldsymbol{x}_j^{k+1}), \tag{7}$$

where $\boldsymbol{y}_i := n\sum_{j\in\mathcal{N}(i)}(\boldsymbol{\nu}_{i,j} - \boldsymbol{\nu}_{j,i})$ explicitly satisfies $\sum_{i=1}^{n}\boldsymbol{y}_i = 0$. Updates (6) and (7) imply optimality and consensus in the optimization process. To see it, let $\boldsymbol{x}_i^\star$ and $\boldsymbol{y}_i^\star$ denotes the fixed points that updates (6) and (7) converge to for any $i \in \{1, \cdots, n\}$. It follows that

- Update (7) implies **consensus**. With $\boldsymbol{x}_i^k \to \boldsymbol{x}_i^\star, \boldsymbol{y}_i^k \to \boldsymbol{y}_i^\star$, we have

$$\boldsymbol{x}_i^\star = (1/|\mathcal{N}(i)|)\sum_{j\in\mathcal{N}(i)}\boldsymbol{x}_j^\star, \quad \forall i \in \{1, \cdots, n\} \implies \boldsymbol{x}_1^\star = \cdots = \boldsymbol{x}_n^\star. \tag{8}$$

- Update (6) implies **optimality**. With the consensus property established in (8), we introduce $\boldsymbol{x}^\star := \boldsymbol{x}_i^\star$. Since $\boldsymbol{x}_i^k \to \boldsymbol{x}^\star, \boldsymbol{y}_i^k \to \boldsymbol{y}_i^\star$ and $\sum_{i=1}^{n}\boldsymbol{y}_i^\star = 0$, we have

$$\nabla f_i(\boldsymbol{x}^\star) + \partial r(\boldsymbol{x}^\star) + \boldsymbol{y}_i^\star \ni 0 \implies \frac{1}{n}\sum_{i=1}^{n}\nabla f_i(\boldsymbol{x}^\star) + \partial r(\boldsymbol{x}^\star) \ni 0, \tag{9}$$

which implies that the consensual fixed point $\boldsymbol{x}^\star$ is the solution to problem (2).

**Mathematics-inspired base update rules.** To learn better update rules than the handcrafted primal-dual updates (6) and (7), we propose the following base parameterized update rules

$$\boldsymbol{x}_i^{k+1} = \boldsymbol{x}_i^k - \boldsymbol{m}_i^k(\nabla f_i(\boldsymbol{x}_i^k), \boldsymbol{g}_i^{k+1}, \boldsymbol{y}_i^k; \boldsymbol{\theta}_{i,1}), \quad \boldsymbol{g}_i^{k+1} \in \partial r(\boldsymbol{x}_i^{k+1}), \tag{10}$$

$$\boldsymbol{y}_i^{k+1} = \boldsymbol{y}_i^k + \boldsymbol{s}_i^k(\{\boldsymbol{x}_i^{k+1} - \boldsymbol{x}_j^{k+1}\}_{j\in\mathcal{N}(i)}; \boldsymbol{\theta}_{i,2}), \tag{11}$$

where $\boldsymbol{m}_i$ and $\boldsymbol{s}_i$ are primal and dual update rules, maintained by node $i$ and parameterized by $\boldsymbol{\theta}_{i,1}$ and $\boldsymbol{\theta}_{i,2}$, respectively. We expect the learned updates (10) and (11) to enforce optimality and consensus when $\boldsymbol{m}_i$ and $\boldsymbol{s}_i$ satisfy certain conditions (see Sec. 3.2 for details). The formats of the inputs to the base update rules in (10) and (11) are inspired by (6) and (7).

However, the above update rules can be further improved. Note that, (10) does not utilize the communicated information when updating the primal variables $\boldsymbol{x}_i$. Intuitively, it is more efficient to use neighbors' information, *i.e.*, $\{\boldsymbol{x}_j, \nabla f_i(\boldsymbol{x}_j), \boldsymbol{g}_j\}_{j\in\mathcal{N}(i)}$ to update both the primal and dual variables. Therefore, we propose

$$\boldsymbol{z}_i^{k+1} = \boldsymbol{x}_i^k - \boldsymbol{m}_i^k(\nabla f_i(\boldsymbol{x}_i^k), \boldsymbol{g}_i^{k+1}, \boldsymbol{y}_i^k; \boldsymbol{\theta}_{i,1}), \quad \boldsymbol{g}_i^{k+1} \in \partial r(\boldsymbol{z}_i^{k+1}), \tag{12}$$

$$\boldsymbol{y}_i^{k+1} = \boldsymbol{y}_i^k + \boldsymbol{s}_i^k(\{\boldsymbol{z}_i^{k+1} - \boldsymbol{z}_j^{k+1}\}_{j\in\mathcal{N}(i)}; \boldsymbol{\theta}_{i,2}), \tag{13}$$

$$\boldsymbol{x}_i^{k+1} = \boldsymbol{z}_i^{k+1} - \boldsymbol{u}_i^k(\{\boldsymbol{z}_i^{k+1} - \boldsymbol{z}_j^{k+1}\}_{j\in\mathcal{N}(i)}; \boldsymbol{\theta}_{i,3}). \tag{14}$$

where $\boldsymbol{z}_i$ is a local auxiliary variable to estimate $\boldsymbol{x}_i$ after one local update (12) within node $i$, and $\boldsymbol{u}_i$ is the newly introduced update rule to update $\boldsymbol{x}_i$ with neighbor's information. Base update rules $\boldsymbol{m}_i, \boldsymbol{s}_i$ and $\boldsymbol{u}_i$ serve as foundations to our MiLoDo framework.

### 3.2 STRUCTURED UPDATE RULES

To ensure the sufficient capacity of update rules $\boldsymbol{m}_i, \boldsymbol{s}_i$ and $\boldsymbol{u}_i$ in practice, one should opt for neural networks to parameterize them. Inspired by the universal approximation theorem, which states that neural networks can approximate any continuous functions, it follows that *searching the parameter space of a neural network model is similar to searching the entire continuous function space*. Therefore, in this subsection, we suppose $\boldsymbol{m}_i, \boldsymbol{s}_i$ and $\boldsymbol{u}_i$ are picked from the following space without specific parameterization.

**Definition 3** (Family of learnable functions): Given a domain $\mathcal{Z}$, we let $\mathrm{J}\boldsymbol{f}(\boldsymbol{z})$ denote the Jacobian matrix of the map $\boldsymbol{f} : \mathcal{Z} \to \mathbb{R}^d$ and $\|\cdot\|_F$ denote the Frobenius norm. We define

$$\mathcal{D}_C(\mathcal{Z}) = \left\{ \boldsymbol{f} : \mathcal{Z} \to \mathbb{R}^d \mid \boldsymbol{f} \text{ is differentiable}, \|\mathrm{J}\boldsymbol{f}(\boldsymbol{z})\|_F \leq C, \forall \boldsymbol{z} \in \mathcal{Z} \right\}$$

as the family of learnable functions.

Specifically, we assume $\boldsymbol{m}_i \in \mathcal{D}_C(\mathbb{R}^{3d})$, and $\boldsymbol{s}_i, \boldsymbol{u}_i \in \mathcal{D}_C(\mathbb{R}^{|\mathcal{N}(i)|d})$. This ensures that our results do not depend on particular parameterizations but rather reflect general principles.

**Mathematical conditions that a good update rule should satisfy.** One may naturally ask: do all the mappings in $\mathcal{D}_C(\mathcal{Z})$ serve as effective rules within the framework of (12)–(14)? If not, can we identify the subset of $\mathcal{D}_C(\mathcal{Z})$ containing desirable update rules by considering fundamental conditions that these rules must fulfill?

Now we examine the mathematical conditions that base update rules (12)–(14) need to satisfy in order to guarantee both consensus and optimality, *i.e.*, $\boldsymbol{x}_1^\star = \cdots = \boldsymbol{x}_n^\star = \boldsymbol{x}^\star$. Inspired by the primal-dual algorithm discussed in Section 3.1, we refer to the base update rules in (12)–(14) as *good ones* if they satisfy the following two conditions.

**Condition 1** (FIXED POINT): For any $\boldsymbol{x}^\star \in \arg\min_{\boldsymbol{x} \in \mathbb{R}^d} f(\boldsymbol{x}) + r(\boldsymbol{x})$, $\boldsymbol{g}_i^\star \in \partial r(\boldsymbol{x}^\star)$ and $\boldsymbol{y}_i^\star = -\nabla f_i(\boldsymbol{x}^\star) - \boldsymbol{g}_i^\star$, it holds for any $i \in \mathcal{V}$ that

$$\lim_{k \to \infty} \boldsymbol{m}_i^k(\nabla f_i(\boldsymbol{x}^\star), \boldsymbol{g}_i^\star, \boldsymbol{y}_i^\star) = \lim_{k \to \infty} \boldsymbol{s}_i^k(\{\boldsymbol{0}_d\}_{j \in \mathcal{N}(i)}) = \lim_{k \to \infty} \boldsymbol{u}_i^k(\{\boldsymbol{0}_d\}_{j \in \mathcal{N}(i)}) = \boldsymbol{0}_d. \tag{15}$$

**Remark.** Here $\boldsymbol{x}^\star$ is an optimal primal solution, and $\boldsymbol{y}_i^\star$ is the dual solution retained at node $i$, according to (9). Condition 1 is derived from a fundamental requirement for a good update rule: if $(\boldsymbol{x}_i^k, \boldsymbol{y}_i^k, \boldsymbol{z}_i^k)$ stay at an optimal solution $(\boldsymbol{x}^\star, \boldsymbol{y}_i^\star, \boldsymbol{x}^\star)$, the next iterate $(\boldsymbol{x}_i^{k+1}, \boldsymbol{y}_i^{k+1}, \boldsymbol{z}_i^{k+1})$ should be fixed. By substituting $\boldsymbol{x}_i^k = \boldsymbol{z}_i^k = \boldsymbol{x}^\star, \boldsymbol{y}_i^k = \boldsymbol{y}_i^\star \in -\nabla f_i(\boldsymbol{x}^\star) - \partial r(\boldsymbol{x}^\star)$ and $\boldsymbol{x}_i^{k+1} = \boldsymbol{z}_i^{k+1} = \boldsymbol{x}^\star, \boldsymbol{y}_i^{k+1} = \boldsymbol{y}_i^\star$ into (12)-(14), we will obtain $\boldsymbol{m}_i^k(\nabla f_i(\boldsymbol{x}^\star), \boldsymbol{g}_i^\star, \boldsymbol{y}_i^\star) = \boldsymbol{0}_d, \boldsymbol{s}_i^k(\{\boldsymbol{0}_d\}_{j \in \mathcal{N}(i)}) = \boldsymbol{0}_d$, and $\boldsymbol{u}_i^k(\{\boldsymbol{0}_d\}_{j \in \mathcal{N}(i)}) = \boldsymbol{0}_d$, and Condition 1 reflects these conditions.

**Condition 2** (GLOBAL CONVERGENCE): For any sequences generated by the base update rules in (12)-(14), there exists $\boldsymbol{x}^\star \in \arg\min_{\boldsymbol{x} \in \mathbb{R}^d} f(\boldsymbol{x}) + r(\boldsymbol{x})$, $\boldsymbol{y}_i^\star = -\nabla f_i(\boldsymbol{x}^\star) - \partial r(\boldsymbol{x}^\star)$ such that

$$\lim_{k \to \infty} \boldsymbol{x}_i^k = \lim_{k \to \infty} \boldsymbol{z}_i^k = \boldsymbol{x}^\star, \quad \lim_{k \to \infty} \boldsymbol{y}_i^k = \boldsymbol{y}_i^\star, \quad \forall i \in \mathcal{V}. \tag{16}$$

**Remark.** With Condition 2, any fixed points of (12)-(14) will be the optimal primal and dual solution to problem (2). This condition enforces both consensus and optimality for update rules in (12)-(14).

**Deriving mathematical structures for base update rules.** The following theorem derives the mathematical structures that the base update rules in (12)–(14) should possess to satisfy the necessary mathematical conditions mentioned above:

**Theorem 1** (MATHEMATICS-INSPIRED STRUCTURE): Given $f_i \in \mathcal{F}_L(\mathbb{R}^d)$, $r \in \mathcal{F}(\mathbb{R}^d)$ and base update rules $\{\boldsymbol{m}_i^k, \boldsymbol{s}_i^k, \boldsymbol{u}_i^k\}_{k=0}^\infty$ with $\boldsymbol{m}_i^k \in \mathcal{D}_C(\mathbb{R}^{3d})$, $\boldsymbol{s}_i^k, \boldsymbol{u}_i^k \in \mathcal{D}_C(\mathbb{R}^{|\mathcal{N}(i)|d})$, if Conditions 1 and 2 hold, there exist $\boldsymbol{P}_i^k, \boldsymbol{P}_{i,j,1}^k, \boldsymbol{P}_{i,j,2}^k \in \mathbb{R}^{d \times d}$ and $\boldsymbol{b}_{i,1}^k, \boldsymbol{b}_{i,2}^k, \boldsymbol{b}_{i,3}^k \in \mathbb{R}^d$ satisfying

$$\boldsymbol{m}_i^k(\nabla f_i(\boldsymbol{x}_i^k), \boldsymbol{g}_i^{k+1}, \boldsymbol{y}_i^k) = \boldsymbol{P}_i^k(\nabla f_i(\boldsymbol{x}_i^k) + \boldsymbol{g}_i^{k+1} + \boldsymbol{y}_i^k) + \boldsymbol{b}_{i,1}^k, \tag{17}$$

$$\boldsymbol{s}_i^k(\{\boldsymbol{z}_i^{k+1} - \boldsymbol{z}_j^{k+1}\}_{j \in \mathcal{N}(i)}) = \sum_{j \in \mathcal{N}(i)} \boldsymbol{P}_{i,j,1}^k(\boldsymbol{z}_i^{k+1} - \boldsymbol{z}_j^{k+1}) + \boldsymbol{b}_{i,2}^k, \tag{18}$$

$$\boldsymbol{u}_i^k(\{\boldsymbol{z}_i^{k+1} - \boldsymbol{z}_j^{k+1}\}_{j \in \mathcal{N}(i)}) = \sum_{j \in \mathcal{N}(i)} \boldsymbol{P}_{i,j,2}^k(\boldsymbol{z}_i^{k+1} - \boldsymbol{z}_j^{k+1}) + \boldsymbol{b}_{i,3}^k, \tag{19}$$

with $\boldsymbol{P}_i^k, \boldsymbol{P}_{i,j,1}^k, \boldsymbol{P}_{i,j,2}^k$ uniformly upper bounded and $\boldsymbol{b}_{i,1}^k, \boldsymbol{b}_{i,2}^k, \boldsymbol{b}_{i,3}^k \to \boldsymbol{0}_d$ as $k \to \infty$. If we further assume $\boldsymbol{P}_i^k$ to be positive definite, base update rule (12) can be uniquely determined through

$$\boldsymbol{z}_i^{k+1} = \mathrm{prox}_{r, \boldsymbol{P}_i^k}\left(\boldsymbol{x}_i^k - \boldsymbol{P}_i^k(\nabla f_i(\boldsymbol{x}_i^k) + \boldsymbol{y}_i^k) - \boldsymbol{b}_{i,1}^k\right), \tag{20}$$

where notation $\mathrm{prox}_{\phi, \boldsymbol{M}}(\boldsymbol{x}) := \arg\min_{\boldsymbol{y}} \phi(\boldsymbol{y}) + \frac{1}{2}\|\boldsymbol{y} - \boldsymbol{x}\|_{\boldsymbol{M}^{-1}}^2$ and $\|\boldsymbol{x}\|_{\boldsymbol{M}} := \sqrt{\boldsymbol{x}^\top \boldsymbol{M} \boldsymbol{x}}$ for positive definite $\boldsymbol{M}$.

**Remark.** Theorem 1 illustrates that the update rules $\boldsymbol{m}_i$, $\boldsymbol{s}_i$, and $\boldsymbol{u}_i$ are not completely free under Conditions 1 and 2. It suggests mathematically-inspired structures for the base update rules, as shown in (17)–(19), where $\boldsymbol{P}_i^k, \boldsymbol{P}_{i,j,1}^k, \boldsymbol{P}_{i,j,2}^k$ can be regarded as preconditioners, while $\boldsymbol{b}_{i,1}^k, \boldsymbol{b}_{i,2}^k, \boldsymbol{b}_{i,3}^k$ are bias terms. We name (17)–(19) as *structured update rules*. As shown in Sec. 5, compared to the base update rules in (12)–(14), these structured update rules benefit from a significantly more condensed parameter space and ensure consensus and optimality upon convergence.

# 4 MiLoDo: An Efficient Math-inspired L2O Framework

Inspired by the structured update rules derived in (17)–(19), this section develops a practical L2O framework that can be used to learn effective decentralized optimizers for solving problem (2).

## 4.1 Making structured update rules efficient to learn

To ensure computational efficiency of the update rules in (17)–(19), we specify those $\boldsymbol{P}$ matrices as diagonal ones. Inspired by Theorem 1, which indicates that the bias terms $\boldsymbol{b}_{i,1}^k, \boldsymbol{b}_{i,2}^k, \boldsymbol{b}_{i,3}^k$ vanish asymptotically, we eliminate these terms. Specifically, we set:

$$\boldsymbol{P}_i^k = \mathrm{Diag}(\boldsymbol{p}_i^k), \ \boldsymbol{P}_{i,j,1}^k = \mathrm{Diag}(\boldsymbol{p}_{i,j,1}^k), \ \boldsymbol{P}_{i,j,2}^k = \mathrm{Diag}(\boldsymbol{p}_{i,j,2}^k), \ \boldsymbol{b}_{i,1}^k = \boldsymbol{b}_{i,2}^k = \boldsymbol{b}_{i,3}^k = \boldsymbol{0}_d, \quad (21)$$

where $\boldsymbol{p}_i^k, \boldsymbol{p}_{i,j,1}^k, \boldsymbol{p}_{i,j,2}^k \in \mathbb{R}_+^d$, and $\boldsymbol{p}_{i,j,1}^k = \boldsymbol{p}_{j,i,1}^k, \forall \{i,j\} \in \mathcal{E}$. With (21), the structured update rules in (17)–(19) can be further simplified as:

$$\boldsymbol{z}_i^{k+1} = \mathrm{prox}_{r, \mathrm{Diag}(\boldsymbol{p}_i^k)} \left( \boldsymbol{x}_i^k - \boldsymbol{p}_i^k \odot (\nabla f_i(\boldsymbol{x}_i^k) + \boldsymbol{y}_i^k) \right), \quad (22)$$

$$\boldsymbol{y}_i^{k+1} = \boldsymbol{y}_i^k + \sum_{j \in \mathcal{N}(i)} \boldsymbol{p}_{i,j,1}^k \odot (\boldsymbol{z}_i^{k+1} - \boldsymbol{z}_j^{k+1}), \quad (23)$$

$$\boldsymbol{x}_i^{k+1} = \boldsymbol{z}_i^{k+1} - \sum_{j \in \mathcal{N}(i)} \boldsymbol{p}_{i,j,2}^k \odot (\boldsymbol{z}_i^{k+1} - \boldsymbol{z}_j^{k+1}). \quad (24)$$

Here, $\odot$ denotes element-wise production. We name (22)–(24) as MiLoDo update rules.

**Remark.** MiLoDo update rules cover state-of-the-art handcrafted decentralized algorithms. If $r \equiv 0$ and we let $\boldsymbol{p}_i^k = \gamma \cdot \boldsymbol{1}_d$, $\boldsymbol{p}_{i,j,1}^k = (w_{ij}/(2\gamma)) \cdot \boldsymbol{1}_d$, $\boldsymbol{p}_{i,j,2}^k = (w_{ij}/2) \cdot \boldsymbol{1}_d$, where $\boldsymbol{1}_d = [1, 1, \cdots, 1]^\top \in \mathbb{R}^d$, MiLoDo update rules reduce to Exact-Diffusion (Yuan et al., 2018b) with symmetric doubly-stochastic gossip matrix $W = (w_{ij})_{n \times n}$ and learning rate $\gamma$. However, MiLoDo update rules are more general than Exact-Diffusion due to the learnable preconditioner $\boldsymbol{p}_i$ and mixing weight $\boldsymbol{p}_{i,j}$.

The following theorem provides theoretical guarantees for MiLoDo update rules, demonstrating that *their fixed points are the primal and dual optimal solutions to problem* (2). To our knowledge, no existing decentralized L2O algorithms could guarantee that their fixed points are optimal solutions.

**Theorem 2** (Exact convergence): Assume $\mathcal{G} = (\mathcal{V}, \mathcal{E})$ is strongly connected, $\{f_i\}_{i \in \mathcal{V}} \subset \mathcal{F}_L(\mathbb{R}^d)$, $r \in \mathcal{F}(\mathbb{R}^d)$ and there exists $0 < m < M < \infty$ such that $[\boldsymbol{p}_i^k]_l \geq m$, $m \leq [\boldsymbol{p}_{i,j,1}^k]_l \leq M$, $|[\boldsymbol{p}_{i,j,2}^k]_l| \leq M$ for all $k \geq 0$ and $1 \leq l \leq d$. Here $[\boldsymbol{x}]_l$ denotes the $l$-th coordinate of the vector $\boldsymbol{x}$. If a sequence generated by (22)-(24) with initialization $\boldsymbol{y}_i^0 = \boldsymbol{0}_d$ converges to $\{\boldsymbol{x}_i^\star, \boldsymbol{y}_i^\star, \boldsymbol{z}_i^\star\}_{i=1}^n$, then this limit must be the primal and dual optimal solutions to problem (2). In other words, there exists $\boldsymbol{x}^\star \in \arg\min_{\boldsymbol{x} \in \mathbb{R}^d} f(\boldsymbol{x}) + r(\boldsymbol{x})$ such that $\boldsymbol{x}_i^\star = \boldsymbol{z}_i^\star = \boldsymbol{x}^\star$ and $\boldsymbol{y}_i^\star \in -\nabla f_i(\boldsymbol{x}^\star) - \partial r(\boldsymbol{x}^\star)$ holds.

## 4.2 LSTM parameterization for MiLoDo update rules

This subsection discusses how to learn $\boldsymbol{p}_i^k$, $\boldsymbol{p}_{i,j,1}^k$ and $\boldsymbol{p}_{i,j,2}^k$ in MiLoDo update rules (22)–(24). To this end, we parameterize $\boldsymbol{p}_i^k$, $\boldsymbol{p}_{i,j,1}^k$ and $\boldsymbol{p}_{i,j,2}^k$ through three local coordinate-wise LSTM neural networks $\phi_{M,i}, \phi_{S,i}, \phi_{U,i}$. Each network is constructed with a single LSTM cell, followed by a 2-layer MLP and an output activation layer. Specifically,

$$\boldsymbol{p}_i^k, \boldsymbol{h}_{M,i}^{k+1} = \phi_{M,i}(\nabla f(\boldsymbol{x}_i^k), \boldsymbol{y}_i^k, \boldsymbol{h}_{M,i}^k; \boldsymbol{\theta}_{M,i}), \quad (25)$$

$$\{\tilde{\boldsymbol{p}}_{i,j,1}^k\}_{j \in \mathcal{N}(i)}, \boldsymbol{h}_{S,i}^{k+1} = \phi_{S,i}(\{\boldsymbol{z}_i^{k+1} - \boldsymbol{z}_j^{k+1}\}_{j \in \mathcal{N}(i)}, \boldsymbol{h}_{S,i}^k; \boldsymbol{\theta}_{S,i}), \quad (26)$$

$$\{\boldsymbol{p}_{i,j,2}^k\}_{j \in \mathcal{N}(i)}, \boldsymbol{h}_{U,i}^{k+1} = \phi_{U,i}(\{\boldsymbol{z}_i^{k+1} - \boldsymbol{z}_j^{k+1}\}_{j \in \mathcal{N}(i)}, \boldsymbol{h}_{U,i}^k; \boldsymbol{\theta}_{U,i}), \quad (27)$$

where $\boldsymbol{h}_{M,i}^k, \boldsymbol{h}_{S,i}^k, \boldsymbol{h}_{U,i}^k$ are hidden states in the LSTM modules with random-initialization, $\boldsymbol{\theta}_{M,i}, \boldsymbol{\theta}_{S,i}, \boldsymbol{\theta}_{U,i}$ are learnable parameters in $\phi_{M,i}, \phi_{S,i}, \phi_{U,i}$, respectively. To achieve $\boldsymbol{p}_{i,j,1}^k = \boldsymbol{p}_{j,i,1}^k$, we compute

$$\boldsymbol{p}_{i,j,1}^k = \boldsymbol{p}_{j,i,1}^k = (\tilde{\boldsymbol{p}}_{i,j,1}^k + \tilde{\boldsymbol{p}}_{j,i,1}^k)/2. \quad (28)$$

---

**Algorithm 1:** MiLoDo Framework

---

**Input:** Optimizee objectives $\{f_i\}_{i \in \mathcal{V}}, r$, network topology $\mathcal{G} = (\mathcal{V}, \mathcal{E})$, LSTM modules
$\quad\quad \{\phi_{M,i}, \phi_{S,i}, \phi_{U,i}\}_{i \in \mathcal{V}}$, number of iterations $K$.

**for** *all nodes $i \in \mathcal{V}$ in parallel* **do**

$\quad$ Initialize variables $\boldsymbol{x}_i^0 = \boldsymbol{y}_i^0 = \boldsymbol{z}_i^0 = \mathbf{0}_d$;

$\quad$ Initialize hidden states $\boldsymbol{h}_{M,i}^0, \boldsymbol{h}_{S,i}^0$ and $\boldsymbol{h}_{P,i}^0$ randomly by normal distribution;

$\quad$ **for** $k = 0, 1, \cdots, K-1$ **do**

$\quad\quad$ Compute $\boldsymbol{p}_i^k$ through (25), update $\boldsymbol{z}_i^{k+1}$ through (22);

$\quad\quad$ Communicate $\boldsymbol{z}_i^{k+1}$ with all neighbors, compute $\tilde{\boldsymbol{p}}_{i,j,1}^k, \boldsymbol{p}_{i,j,2}^k$ through (26)(27);

$\quad\quad$ **for** $j \in \mathcal{N}(i)$ **do**

$\quad\quad\quad$ Send $\tilde{\boldsymbol{p}}_{i,j,1}^k$ to and receive $\tilde{\boldsymbol{p}}_{j,i,1}^k$ from node $j$;

$\quad\quad$ Compute $\boldsymbol{p}_{i,j,1}^k$ through (28), update $\boldsymbol{y}_i^{k+1}, \boldsymbol{x}_i^{k+1}$ through (23)(24).

---

Combining the structured update rules with LSTM parameterization, we obtain the complete architecture of MiLoDo framework, as illustrated in Algorithm 1.

## 4.3 TRAINING MiLoDo FRAMEWORK

To determine $\boldsymbol{\Theta} = \{\boldsymbol{\theta}_{M,i}, \boldsymbol{\theta}_{S,i}, \boldsymbol{\theta}_{U,i}\}_{i=1}^n$ in (25) – (27), we evaluate and refine the performance of the optimizer over the initial $K$ steps on a batch of training optimizees $\mathcal{F}_B$, *i.e.*,

$$\min_{\boldsymbol{\Theta}} \mathcal{L}_K(\boldsymbol{\Theta}, \mathcal{F}_B) := \frac{1}{|\mathcal{F}_B|} \sum_{f \in \mathcal{F}_B} \left[ \frac{1}{K} \sum_{k=1}^K f(\bar{\boldsymbol{x}}^k) \right]. \tag{29}$$

The variable $\bar{\boldsymbol{x}}^k = \frac{1}{n} \sum_{i=1}^n \boldsymbol{x}_i^k$ is the average of local variables $\boldsymbol{x}_i$'s. Typically, we set $K = 100$ and train the model by truncated Back Propagation Through Time (BPTT) with a truncation length of $K_T = 20$, following a common setup in previous L2O approaches (Chen et al., 2017; Lv et al., 2017; Wichrowska et al., 2017; Metz et al., 2019; Cao et al., 2019; Chen et al., 2020c;b). *More specifically, we divide the $K$ iterations into $K/K_T$ segments of length $K_T$ and train the optimizer on them separately. Such a training strategy is denoted by $(K_T, K) = (20, 100)$ throughout this paper.* More training techniques such as initialization and multi-stage training are in Sec. E.1.

## 5 EXPERIMENTAL RESULTS

This section presents numerical experiments to validate the strong generalization capability of the MiLoDo-trained optimizer to out-of-distribution tasks. Additionally, we compare it with state-of-the-art handcrafted optimizers such as Prox-DGD, PG-EXTRA, Prox-ATC, Prox-ED, DAPG (Ye et al., 2020), ODAPG (Ye & Chang, 2023), as well as the learned optimizer LPG-EXTRA (Wang et al., 2021). Note that LPG-EXTRA, an algorithm unrolling method, is confined to solving unseen problems in the test dataset for a maximum of 100 iterations due to the memory bottleneck imposed by its unrolling structure. Conversely, all handcrafted and MiLoDo-trained optimizers can be tested over much longer horizons, typically in the order of $10^5$.

**Experimental setup.** In our experiments, we use a special initialization and a multi-stage training strategy discussed in Sec. E.1. Specifically, we train MiLoDo in five stages with $(K_T, K) = (5, 10)$, $(10, 20)$, $(20, 40)$, $(40, 80)$, and $(20, 100)$, using Adam with learning rates of $5 \times 10^{-4}, 1 \times 10^{-4}, 5 \times 10^{-5}, 1 \times 10^{-5}$, and $1 \times 10^{-5}$, for 20, 10, 10, 10, and 5 epochs, respectively. Throughout all stages, the Adam optimizer is configured with momentum parameters $(\beta_1, \beta_2) = (0.9, 0.999)$ and the batch size is set at 32. More data collection/generation and training details can be found in Appendix E.

**Target problems.** Our target problems include LASSO, logistic regression, MLP and ResNet. In all experiments, we use the shape $(n, d, N, \lambda)$ to represent different characteristics of the optimizees, where $n$ represents the number of nodes in the decentralized network, $d$ represents the feature dimension, $N$ represents the number of data samples held by each worker, and $\lambda$ represents the $\ell_1$ regularization coefficient. Without further clarification, we consider a ring topology for the network.

**Training sets.** MiLoDo optimizers in this section are trained on two different training sets: specialized and meta training set. Specifically, the *specialized training set* consists of 512 synthetic

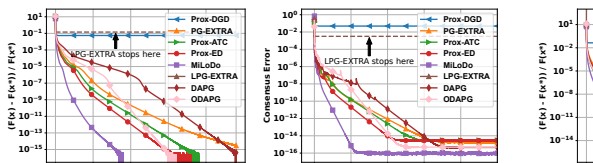

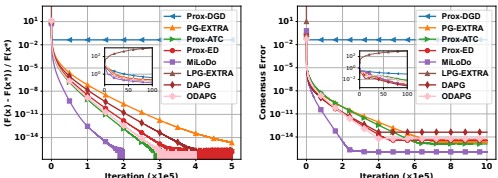

Figure 2: MiLoDo-optimizer trained on synthetic LASSO$(10, 300, 10, 0.1)$ and tested on unseen LASSO$(10, 300, 10, 0.1)$ instances.

Figure 3: MiLoDo-optimizer trained on synthetic LASSO$(10, 300, 10, 0.1)$ and tested on synthetic LASSO$(10, 30000, 1000, 0.1)$.

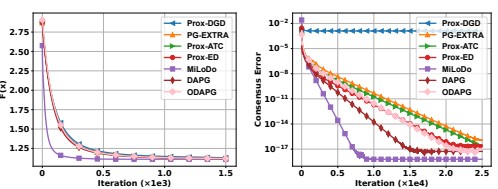

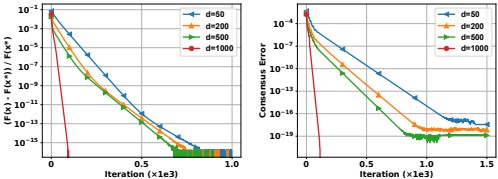

Figure 4: MiLoDo-optimizer trained on meta training set and tested on LASSO$(10, 200, 10, 0.05)$ with real dataset BSDS500(Martin et al., 2001).

Figure 5: MiLoDo-optimizer trained on meta training set and tested on Logistic$(10, d, 100, 0.1)$ with $d \in \{50, 200, 500, 1000\}$.

LASSO$(10, 300, 10, 0.1)$ instances, while the *meta training set* consists of 1280 synthetic instances with various sizes, including 64 LASSO$(10, 500, N, 0.1)$ for each $N \in \{5, 10, 15, \cdots, 100\}$.

**Generalization to longer testing iterations.** MiLoDo optimizer, trained to operate for a small number of iterations with training problem sets, performs well for significantly more iterations when solving unseen problem sets. As illustrated in Fig. 2, MiLoDo trained on LASSO$(10, 300, 10, 0.1)$ with $(K_T, K) = (20, 100)$, performs robustly for up to 100,000 testing iterations on unseen LASSO$(10, 300, 10, 0.1)$ instances. In contrast, the other learned optimizer, LPG-EXTRA, can only be applied for 100 iterations limited by its memory bottleneck. Moreover, compared with handcrafted optimizers, MiLoDo achieves a $1.7\times$ speedup in convergence and more than a $2\times$ speedup in consensus.

**Generalization to higher problem dimensions.** MiLoDo optimizer trained with low-dimensional problems can be generalized to solve problems with much higher dimensions. As illustrated in Fig. 3, MiLoDo trained on LASSO$(10, 300, 10, 0.1)$ with a problem dimension 300 performs consistently well on LASSO$(10, 30000, 1000, 0.1)$ instances with a much higher dimension of 30,000.

**Generalization to real data distributions.** MiLoDo optimizer trained with meta training dataset (synthetic LASSO) can be generalized to real data distributions. As illustrated in Fig. 4, MiLoDo trained on the meta training set performs consistently well on LASSO$(10, 200, 10, 0.05)$ constructed with real dataset BSDS500 (Martin et al., 2001), achieving more than a $2.5\times$ speedup in both convergence and consensus rate.

**Generalization to different problem types.** MiLoDo optimizer trained with meta-training set can generalize to different problem types. As depicted in Fig. 5, MiLoDo trained on the meta-training set, which consists solely of LASSO problems, converges precisely to the global solutions of unseen logistic regression problems with varying feature dimensions $d \in \{50, 200, 500, 1000\}$.

**Efficacy in neural network training scenarios.** The efficacy of MiLoDo extends to the realm of neural network training, a domain characterized by high computational complexity and strong non-convexity. As shown in Fig. 6, MiLoDo consistently achieves a $2\times$ speedup in training MLP on the MNIST (Deng, 2012) dataset, compared to other baseline methods. MiLoDo also achieves a $2\times$ speedup in training ResNet on the CIFAR-10 (Krizhevsky, 2009) dataset, as illustrated in Fig. 7. This performance underscores MiLoDo's ability to efficiently navigate neural networks' complex loss landscapes, significantly enhancing distributed deep learning.

**Scalability to more complex topologies and larger networks.** MiLoDo optimizer consistently performs well on complex and large-scale networks, showcasing its superior scalability. As shown in Fig.8, MiLoDo consistently enhances efficiency on more complex network topologies and larger networks, achieving a $1.5\times$ speedup on an exponential graph topology, and a $3\times$ speedup on a

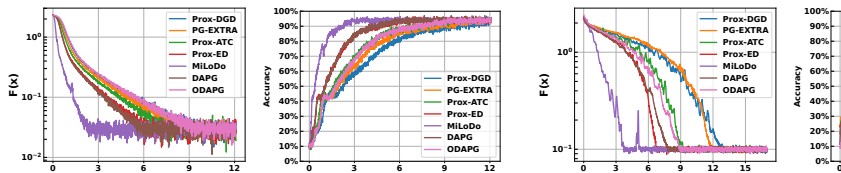

Figure 6: MiLoDo-optimizer trained on MLP(10, 13002, 1000, 0) with MNIST dataset and tested on MLP(10, 13002, 5000, 0).

Figure 7: MiLoDo-optimizer trained on ResNet (5, 78042, 500, 0) with CIFAR-10 dataset and tested on ResNet(5, 78042, 5000, 0).

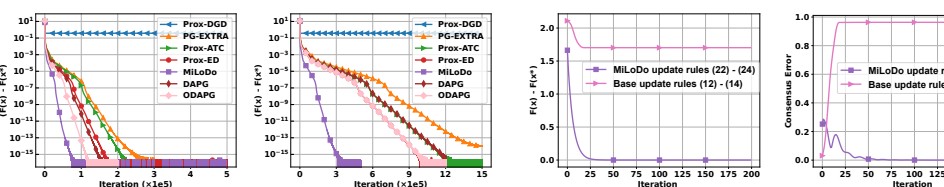

Figure 8: MiLoDo-optimizer trained on different network settings: (left) an exponential graph topology of 10 nodes, (right) a network of 50 nodes.

Figure 9: Testing trained optimizer using base update rules and structured update rules on synthetic LASSO(10, 10, 5, 0).

50-node network. Further tests confirm its effectiveness across various topologies and a 100-node network. Detailed results are in Appendix E.4.

**Influence of math-inspired structures.** The mathematics-inspired structures are crucial for the success of MiLoDo. To see this, we directly parameterize the base update rules (12)-(14) targeting a smooth problem LASSO(10, 10, 5, 0). Detailed experimental setups are deferred to Appendix E.5. As illustrated in Fig. 9, directly parameterizing the base update rules fails to learn good optimizers.

**Runtime studies.** When compared to handcrafted optimizers like Prox-ED, MiLoDo faces a higher per-iteration computational cost due to additional neural network calculations for $\{\boldsymbol{p}_i^k, \boldsymbol{p}_{i,j,1}^k, \boldsymbol{p}_{i,j,2}^k\}$. To assess whether its benefits outweigh these costs, we compre its running time with Prox-ED. As illustrated in Table 1, MiLoDo exhibits only a slight increase in computational cost, about 18.4%, while achieving a significant convergence speedup, resulting in a $1.6\times \sim 2.1\times$ speedup in total.

Table 1: Runtime Comparison on LASSO(10, 30000, 1000, 0.1).

|  | Time/Iters | Stopping condition: Gap $< 10^{-7}$ | | Stopping condition: Gap $< 10^{-15}$ | |
|---|---|---|---|---|---|
|  |  | Iters | Total Time | Iters | Total Time |
| MiLoDo | 5.91 ms | 2.45e+04 | 144.80 s | 1.62e+05 | 957.42 s |
| Prox-ED | 4.99 ms | 6.22e+04 | 310.38 s | 3.07e+05 | 1531.93 s |

**More experimental results.** More testing results on MiLoDo optimizers trained with meta training set and Logistic regression optimizees are in Appendix E.4. We also conduct ablation studies on the mixing matrices, see Appendix E.5.

## 6 CONCLUSIONS AND LIMITATIONS

We propose MiLoDo, a mathematics-inspired L2O framework for decentralized optimization. With its mathematics-inspired structure, the MiLoDo-trained optimizer can generalize to tasks with varying data distributions, problem types, and feature dimensions. Moreover, MiLoDo-trained optimizer outperforms handcrafted optimizers in convergence rate. MiLoDo currently learns separate parameters specifically for each neighbor relationship (i.e., communication links), but training a set of shared parameters across all nodes could simplify the framework. This would align MiLoDo with message-passing graph neural networks, which are more scalable and inherently permutation-equivariant, making them well-suited for graph-based problems. This presents a promising future direction.

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

# APPENDIX

## CONTENTS

## A   MORE RELATED WORK

**Learning to optimize.** The concept of L2O dates back to the 1990s (Thrun & Pratt, 1998). Different L2O approaches exist: Plug-and-Play (PnP) (Venkatakrishnan et al., 2013) approximates expensive functions in traditional algorithms; algorithm unrolling (Gregor & LeCun, 2010; Moreau & Bruna, 2017; Chen et al., 2018; Liu & Chen, 2019; Ito et al., 2019; Yang et al., 2016; Zhang & Ghanem, 2018; Adler & Öktem, 2018; Solomon et al., 2019) models the entire procedure as a neural network, effective for domains like image/signal processing; generic L2O (Andrychowicz et al., 2016; Lv et al., 2017; Wichrowska et al., 2017; Wu et al., 2018; Metz et al., 2019; Chen et al., 2020a; Shen et al., 2020; Harrison et al., 2022; Micaelli & Storkey, 2021; Metz et al., 2018; 2022a;b; Jain et al., 2023; Liu et al., 2023), which is more related to this paper, parameterizes update rules using current states, enabling flexibility across applications. Some other studies also atempts to learn machine learning models to accelerate the discrete problems solving(Bengio et al., 2021).

**Decentralized optimization.** Decentralized optimization has been extensively studied, dating back to early algorithms like decentralized gradient descent (DGD) (Nedic & Ozdaglar, 2009; Yuan et al., 2016), Diffusion (Lopes & Sayed, 2008; Chen & Sayed, 2012), and dual averaging (Duchi et al., 2011) from the signal processing and control communities. These were followed by primal-dual methods such as ADMM variants (Shi et al., 2014), explicit bias-correction techniques (Shi et al., 2015a; Yuan et al., 2018b; Li et al., 2019), Gradient-Tracking (Xu et al., 2015; Di Lorenzo & Scutari, 2016; Nedic et al., 2017). More recently, decentralized stochastic gradient descent (DSGD) (Lian et al., 2017) has gained significant attentions in deep learning. For non-smooth optimization problems, effective algorithms like PG-EXTRA (Shi et al., 2015b), PG-Exact-Diffusion (Yuan et al., 2018a), and PG-Gradient-Tracking (Alghunaim et al., 2020) utilize the proximal gradient method to solve them. A unified decentralized framework is developed in Alghunaim et al. (2020) and Xu et al. (2021) to unify various decentralized algorithms. All these algorithms are driven by expert knowledge.

# B MISSING PROOFS

## B.1 PRELIMINARIES

**Lemma 1** (Liu et al. (2023), Lemma 1)**:** For any operator $\mathbf{o} \in \mathcal{D}_C(\mathbb{R}^{m \times n})$ and any $\mathbf{x}_1, \mathbf{y}_1, \cdots \mathbf{x}_m,$ $\mathbf{y}_m \in \mathbb{R}^n$, there exists matrices $\mathbf{J}_1, \mathbf{J}_2, \cdots, \mathbf{J}_m \in \mathbb{R}^{n \times n}$ such that

$$\mathbf{o}(\mathbf{x}_1, \cdots, \mathbf{x}_m) - \mathbf{o}(\mathbf{y}_1, \cdots, \mathbf{y}_m) = \sum_{j=1}^{m} \mathbf{J}_j (\mathbf{x}_j - \mathbf{y}_j),$$

and $\|\mathbf{J}_1\|_F \leq \sqrt{n}C, \cdots, \|\mathbf{J}_m\|_F \leq \sqrt{n}C$.

This lemma is an extension of the mean value theorem.

## B.2 PROOF OF THEOREM 1

*Proof.* By Lemma 1, there exists $\boldsymbol{Q}_{i,1}^k, \boldsymbol{Q}_{i,2}^k, \boldsymbol{Q}_{i,3}^k \in \mathbb{R}^{d \times d}$ such that $\|\boldsymbol{Q}_{i,1}^k\|_F \leq \sqrt{d}C, \|\boldsymbol{Q}_{i,2}^k\|_F \leq \sqrt{d}C, \|\boldsymbol{Q}_{i,3}^k\|_F \leq \sqrt{d}C$ and

$$
\begin{aligned}
\boldsymbol{m}_i^k(\nabla f_i(\boldsymbol{x}_i^k), \boldsymbol{g}_i^{k+1}, \boldsymbol{y}_i^k) =& \boldsymbol{m}_i^k(\nabla f_i(\boldsymbol{x}^\star), -\nabla f_i(\boldsymbol{x}^\star) - \boldsymbol{y}_i^\star, \boldsymbol{y}_i^\star) + \boldsymbol{Q}_{i,1}^k(\nabla f_i(\boldsymbol{x}_i^k) - \nabla f_i(\boldsymbol{x}^\star)) \\
& + \boldsymbol{Q}_{i,2}^k(\boldsymbol{g}_i^{k+1} + \nabla f_i(\boldsymbol{x}^\star) + \boldsymbol{y}_i^\star) + \boldsymbol{Q}_{i,3}^k(\boldsymbol{y}_i^k - \boldsymbol{y}_i^\star) \\
=& \boldsymbol{Q}_{i,2}^k(\nabla f_i(\boldsymbol{x}_i^k) + \boldsymbol{g}_i^{k+1} + \boldsymbol{y}_i^k) + (\boldsymbol{Q}_{i,1}^k - \boldsymbol{Q}_{i,2}^k)(\nabla f_i(\boldsymbol{x}_i^k) - \nabla f_i(\boldsymbol{x}^\star)) \\
& + (\boldsymbol{Q}_{i,3}^k - \boldsymbol{Q}_{i,2}^k)(\boldsymbol{y}_i^k - \boldsymbol{y}_i^\star) + \boldsymbol{m}_i^k(\nabla f_i(\boldsymbol{x}^\star), -\nabla f_i(\boldsymbol{x}^\star) - \boldsymbol{y}_i^\star, \boldsymbol{y}_i^\star).
\end{aligned}
$$

Letting

$$\boldsymbol{P}_i^k = \boldsymbol{Q}_{i,2}^k$$

and

$$
\begin{aligned}
\boldsymbol{b}_{i,1}^k =& (\boldsymbol{Q}_{i,1}^k - \boldsymbol{Q}_{i,2}^k)(\nabla f_i(\boldsymbol{x}_i^k) - \nabla f_i(\boldsymbol{x}^\star)) + (\boldsymbol{Q}_{i,3}^k - \boldsymbol{Q}_{i,2}^k)(\boldsymbol{y}_i^k - \boldsymbol{y}_i^\star) \\
& + \boldsymbol{m}_i^k(\nabla f_i(\boldsymbol{x}^\star), -\nabla f_i(\boldsymbol{x}^\star) - \boldsymbol{y}_i^\star, \boldsymbol{y}_i^\star),
\end{aligned}
$$

the above equation can be reorganized into (17). Addtionally, we have $\|\boldsymbol{P}_i^k\|_F \leq \sqrt{d}C$ and $\boldsymbol{b}_{i,1}^k \to \boldsymbol{0}_d$ thanks to Condition 1. Similarly, there exists $\boldsymbol{Q}_{i,j,1}^k, \boldsymbol{Q}_{i,j,2}^k \in \mathbb{R}^{d \times d}$ such that $\|\boldsymbol{Q}_{i,j,1}^k\|_F \leq \sqrt{d}C,$ $\|\boldsymbol{Q}_{i,j,2}^k\|_F \leq \sqrt{d}C$ and

$$\boldsymbol{s}_i^k(\{\boldsymbol{z}_i^{k+1} - \boldsymbol{z}_j^{k+1}\}_{j \in \mathcal{N}(i)}) = \boldsymbol{s}_i^k(\{\boldsymbol{0}_d\}_{j \in \mathcal{N}(i)}) + \sum_{j \in \mathcal{N}(i)} \boldsymbol{Q}_{i,j,1}^k(\boldsymbol{z}_i^{k+1} - \boldsymbol{z}_j^{k+1}),$$

$$\boldsymbol{u}_i^k(\{\boldsymbol{z}_i^{k+1} - \boldsymbol{z}_j^{k+1}\}_{j \in \mathcal{N}(i)}) = \boldsymbol{u}_i^k(\{\boldsymbol{0}_d\}_{j \in \mathcal{N}(i)}) + \sum_{j \in \mathcal{N}(i)} \boldsymbol{Q}_{i,j,2}^k(\boldsymbol{z}_i^{k+1} - \boldsymbol{z}_j^{k+1}).$$

Thus, it is sufficient to obtain (18) and (19) by letting $\boldsymbol{P}_{i,j,1}^k = \boldsymbol{Q}_{i,j,1}^k$, $\boldsymbol{P}_{i,j,2}^k = \boldsymbol{Q}_{i,j,2}^k$, $\boldsymbol{b}_{i,2}^k = \boldsymbol{s}_i^k(\{\boldsymbol{0}_d\}_{j \in \mathcal{N}(i)})$, and $\boldsymbol{b}_{i,3}^k = \boldsymbol{u}_i^k(\{\boldsymbol{0}_d\}_{j \in \mathcal{N}(i)})$.

Now we rewrite (12) as

$$\boldsymbol{z}_i^{k+1} = \boldsymbol{x}_i^k - \boldsymbol{P}_i^k(\nabla f_i(\boldsymbol{x}_i^k) + \boldsymbol{g}_i^{k+1} + \boldsymbol{y}_i^k) - \boldsymbol{b}_{i,1}^k, \quad \boldsymbol{g}_i^{k+1} \in \partial r(\boldsymbol{z}_i^{k+1}). \tag{30}$$

If we further assume $\boldsymbol{P}_i^k$ to be symmetric positive definite, (30) implies

$$\boldsymbol{0}_d \in \partial r(\boldsymbol{z}_i^{k+1}) + (\boldsymbol{P}_i^k)^{-1} \left( \boldsymbol{z}_i^{k+1} - \boldsymbol{x}_i^k + \boldsymbol{b}_{i,1}^k + \boldsymbol{P}_i^k(\nabla f_i(\boldsymbol{x}_i^k) + \boldsymbol{y}_i^k) \right).$$

Consequently, $\boldsymbol{z}_i^{k+1}$ coincides with the unique solution to the following strongly-convex optimization problem:

$$\min_{\boldsymbol{x} \in \mathbb{R}^d} \quad r(\boldsymbol{x}) + \frac{1}{2} \|\boldsymbol{x} - (\boldsymbol{x}_i^k - \boldsymbol{P}_i^k(\nabla f_i(\boldsymbol{x}_i^k) + \boldsymbol{y}_i^k) - \boldsymbol{b}_{i,1}^k)\|_{(\boldsymbol{P}_i^k)^{-1}}^2,$$

*i.e.,*

$$\boldsymbol{z}_i^{k+1} = \text{prox}_{r, \boldsymbol{P}_i^k}(\boldsymbol{x}_i^k - \boldsymbol{P}_i^k(\nabla f_i(\boldsymbol{x}_i^k) + \boldsymbol{y}_i^k) - \boldsymbol{b}_{i,1}^k),$$

which finishes the proof. $\qquad\square$

### B.3 PROOF OF THEOREM 2

*Proof.* Let $\mathcal{I}_l = \{i \in \mathcal{V} \mid [z_i^\star]_l = \max_{j \in \mathcal{V}}[z_j^\star]_l\}$ be the set of indices $i$'s with the largest $[z_i^\star]_l$'s. We first prove the following statement:

$$i \in \mathcal{I}_l \Rightarrow \mathcal{N}(i) \subset \mathcal{I}_l. \tag{31}$$

Suppose there exists $i \in \mathcal{I}_l$ and $j \in \mathcal{N}(i) \backslash \mathcal{I}_l$, it holds that $[z_i^\star]_l > [z_j^\star]_l$. Define $\delta = [z_i^\star]_l - [z_j^\star]_l > 0$ and choose an $\epsilon \in \left(0, \frac{m\delta}{2(nM+1)}\right)$. By convergence property there exists $K \geq 0$ such that for any $k \geq K$,

$$|[x_i^k]_l - [x_i^\star]_l| \leq \epsilon, \quad |[y_i^k]_l - [y_i^\star]_l| \leq \epsilon, \quad |[z_i^k]_l - [z_i^\star]_l| \leq \epsilon, \quad \forall i \in \mathcal{V}, 1 \leq l \leq d.$$

By iteration step (23) we have

$$[y_i^{k+1}]_l = [y_i^k]_l + \sum_{j \in \mathcal{N}(i)} [p_{i,j,1}^k]_l([z_i^{k+1}]_l - [z_j^{k+1}]_l).$$

Fix $k \geq K$ and let $\mathcal{N}_1 = \{j \in \mathcal{N}(i) | [z_i^{k+1}]_l \geq [z_j^{k+1}]_l\}$, $\mathcal{N}_2 = \{j \in \mathcal{N}(i) | [z_i^{k+1}]_l < [z_j^{k+1}]_l\}$, we have

$$m(\delta - 2\epsilon) \leq [p_{i,j,1}^k]_l([z_i^{k+1}]_l - [z_j^{k+1}]_l) \leq \sum_{\tau \in \mathcal{N}_1} [p_{i,\tau,1}^k]_l([z_i^{k+1}]_l - [z_\tau^{k+1}]_l)$$

$$= [y_i^{k+1}]_l - [y_i^k]_l - \sum_{\tau \in \mathcal{N}_2} [p_{i,\tau,1}^k]_l([z_i^{k+1}]_l - [z_\tau^{k+1}]_l)$$

$$\leq 2\epsilon + M(n-1) \cdot (2\epsilon),$$

which implies

$$m\delta \leq 2(nM+1)\epsilon,$$

a contradiction. Consequently, (31) holds and thus together with the strongly connectivity we obtain $\mathcal{I}_l = \mathcal{V}$, i.e., $[z_1^\star]_l = [z_2^\star]_l = \cdots = [z_n^\star]_l$. By arbitrariness of $l$, we conclude that there exists $x^\star \in \mathbb{R}^d$ such that $z_i^\star = x^\star$ for any $i \in \mathcal{V}$. By iteration step (24) and $|[p_{i,j,2}^k]_l| \leq M$ we have

$$x_i^\star = \lim_{k \to \infty} x_i^{k+1} = \lim_{k \to \infty} \left( z_i^{k+1} - \sum_{j \in \mathcal{N}(i)} p_{i,j,2}^k \odot (z_i^{k+1} - z_j^{k+1}) \right) = x^\star, \quad \forall i \in \mathcal{V}.$$

By iteration step (22), we have

$$z_i^{k+1} = \text{prox}_{r, \text{Diag}(p_i^k)} \left( x_i^k - p_i^k \odot (\nabla f_i(x_i^k) + y_i^k) \right)$$

$$= \arg\min_{x \in \mathbb{R}^d} r(x) + \frac{1}{2}\|x - x_i^k + p_i^k \odot (\nabla f_i(x_i^k) + y_i^k)\|_{(\text{Diag}(p_i))^{-1}}^2,$$

which is equivalent to

$$0 \in \partial r(z_i^{k+1}) + (\text{Diag}(p_i))^{-1}\left(z_i^{k+1} - x_i^k + p_i^k \odot (\nabla f_i(x_i^k) + y_i^k)\right)$$

$$\Leftrightarrow -\nabla f_i(x_i^k) - y_i^k - (\text{Diag}(p_i))^{-1}(z_i^{k+1} - x_i^k) \in \partial r(z_i^{k+1}).$$

Denote $g_i^{k+1} = -\nabla f_i(x_i^k) - y_i^k - (\text{Diag}(p_i))^{-1}(z_i^{k+1} - x_i^k) \in \partial r(z_i^{k+1})$, we have

$$r(x) \geq r(z_i^{k+1}) + \langle g_i^{k+1}, x - z_i^{k+1}\rangle, \quad \forall x \in \mathbb{R}^d. \tag{32}$$

Since $r \in \mathcal{F}(\mathbb{R}^d)$ inherits lower semi-continuity, and $0 < [p_i^k]_l^{-1} \leq 1/m$, (32) implies

$$r(x) \geq \liminf_{k \to \infty} r(z_i^{k+1}) + \lim_{k \to \infty} \langle g_i^{k+1}, x - z_i^{k+1}\rangle$$

$$\geq r(x^\star) + \langle -\nabla f_i(x^\star) - y_i^\star, x - x^\star\rangle, \quad \forall x \in \mathbb{R}^d.$$

As a result, $g_i^\star := -\nabla f_i(x^\star) - y_i^\star \in \partial r(x^\star)$. The last thing is to show $x^\star \in \arg\min_{x \in \mathbb{R}^d} f(x) + r(x)$. Adding (23) for all $i \in \mathcal{V}$, we have

$$\sum_{i \in \mathcal{V}} y_i^{k+1} = \sum_{i \in \mathcal{V}} \left( y_i^k + \sum_{j \in \mathcal{N}(i)} p_{i,j,1}^k \odot (z_i^{k+1} - z_j^{k+1}) \right)$$

$$= \sum_{i \in \mathcal{V}} \boldsymbol{y}_i^k + \sum_{\{i,j\} \in \mathcal{E}} (\boldsymbol{p}_{i,j,1}^k - \boldsymbol{p}_{j,i,1}^k) \odot (\boldsymbol{z}_i^{k+1} - \boldsymbol{z}_j^{k+1})$$

$$= \sum_{i \in \mathcal{V}} \boldsymbol{y}_i^k, \quad \forall k \geq 0, \tag{33}$$

where the last equality uses $\boldsymbol{p}_{i,j,1}^k = \boldsymbol{p}_{j,i,1}^k$. By initialization $\boldsymbol{y}_i^0 = \mathbf{0}_d$, (33) implies

$$\sum_{i \in \mathcal{V}} \boldsymbol{y}_i^\star = \lim_{k \to \infty} \sum_{i \in \mathcal{V}} \boldsymbol{y}_i^k = \lim_{k \to \infty} \mathbf{0}_d = \mathbf{0}_d,$$

thus

$$\partial r(\boldsymbol{x}^\star) \ni \frac{1}{n} \sum_{i=1}^n \boldsymbol{g}_i^\star = \frac{1}{n} \sum_{i=1}^n -\nabla f_i(\boldsymbol{x}^\star) - \boldsymbol{y}_i^\star = -\nabla f(\boldsymbol{x}^\star),$$

which is exactly $\boldsymbol{x}^\star \in \arg\min_{\boldsymbol{x} \in \mathbb{R}^d} f(\boldsymbol{x}) + r(\boldsymbol{x})$. $\qquad\square$

## C ILLUSTRATION OF MILODO FRAMEWORK

To better understanding the two components, *i.e.*, the MiLoDo update rules (22)-(24) and the LSTM parameterization (25)-(27) and how they make up the whole MiLoDo optimizer, we illustrate the interaction beween them in Fig. 10.

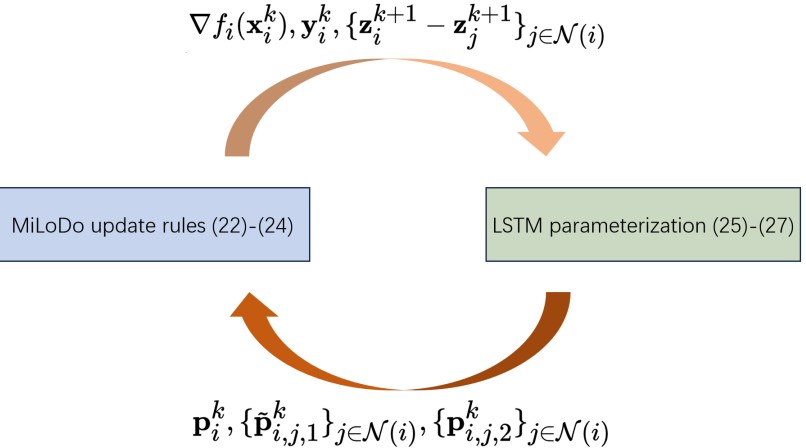

Figure 10: The interaction between MiLoDo update rules (22)-(24) and the LSTM parameterization (25)-(27).

## D ROBUST IMPLEMENTATION OF DECENTRALIZED ALGORITHMS

**Common implementation.** Traditional decentralized algorithms like Prox-ED, PG-EXTRA, Prox-ATC use a doubly-stochastic gossip matrix $W$ for information aggregation. A common implementation of the aggregation step $\tilde{\boldsymbol{X}} = W\boldsymbol{X}$ is to compute

$$\tilde{\boldsymbol{x}}_i = \sum_{j \in \mathcal{N}(i)} w_{ij} \boldsymbol{x}_j \tag{34}$$

on each node $i$. However, (34) is not a robust implementation. As illustrated in Fig. 11, using (34) and the same hyperparameter settings, Prox-ED with FP32 fails to converge to the desired precision while that with FP64 succeeds.

**Robust implementation.** When represented with FP32, the elements in matrix $W$ tend to have bigger noise, which largely violates the row-stochastic property. Continually applying such an inexact

estimation of $W$ is the major reason behind the failure of the common implementation. This motivates us to consider the following equivalent implementation:

$$\tilde{\boldsymbol{x}}_i = \boldsymbol{x}_i - \sum_{j \in \mathcal{N}(i)} w_{ij}(\boldsymbol{x}_i - \boldsymbol{x}_j). \tag{35}$$

Implementation (35) is more robust as it maintains the row-stochastic property of the gossip matrix no matter how much noise is added to $W$ by the low presentation precision. As illustrated in Fig. 11, Prox-ED with robust implementation successfully converges to the desired precision under the same hyperparameter settings.

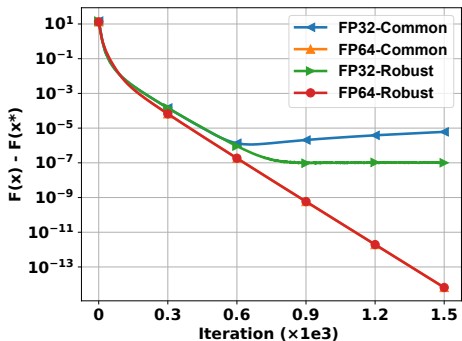

Figure 11: Testing results on synthetic LASSO$(10, 50, 10, 0)$ of Prox-ED with different implementation under varying precision settings. 'Common' and 'Robust' represent common implementation (34) and robust implementation (35), respectively.

**Robustness of MiLoDo.** It's worth noting that, the structured update rules of MiLoDo does not depend on the doubly-stochastic matrix $W$. The utilization of term $\boldsymbol{z}_i^k - \boldsymbol{z}_j^k$ when aggregating neighboring information in update rules of MiLoDo is also similar to the robust implementation (35). In practice, we observe that applying FP32 in our experiments does not affect the exact convergence of MiLoDo-trained optimizers.

# E  EXPERIMENTAL SPECIFICATIONS

## E.1  TRAINING STRATEGIES

**Initialization strategies.** We consider two initialization strategies for MiLoDo training: *random* and *special* initialization. In random initialization, all learnable parameters $\{\boldsymbol{\theta}_{M,i}, \boldsymbol{\theta}_{S,i}, \boldsymbol{\theta}_{U,i}\}_{i \in \mathcal{V}}$ are randomly initialized using PyTorch defaults. In special initialization, parameters are initialized to mimic traditional decentralized algorithms by setting the weights of the final affine layers to zero and biases to desired output values. Specifically, given the gossip matrix $W = (w_{ij})_{n \times n}$ and learning rate $\gamma$ utilized in Exact-Diffusion, biases in the final affine layers for $\boldsymbol{p}_i^k, \tilde{\boldsymbol{p}}_{i,j,1}^k, \boldsymbol{p}_{i,j,2}^k$ are initialized as $\gamma, \ln(w_{ij}/(2\gamma)), w_{ij}/2$, respectively. Applying $\ln(\cdot)$ accounts for Exponential activation.

We would like to remark that each of the two initialization strategies has its pros and cons. With random initialization, the objective function value is likely to blow up quickly, leading to excessively large gradients or meaningless values (e.g., inf/NaNs). With special initialization, it may be too close to local minima, such that MiLoDo might not gain enough advantage over handcrafted algorithms.

**Multi-stage training.** As discussed above, random initialization of the MiLoDo optimizer results in numerical instability during training. To address this issue, we initially teach the model to optimize within a few iterations by using a short training length such as $(K_T, K) = (5, 10)$. As the model starts exhibiting desired behaviors, such as $\sum_{k=1}^{5} f(\bar{\boldsymbol{x}}^k) > \sum_{k=6}^{10} f(\bar{\boldsymbol{x}}^k)$, we progressively increase the training length. This iterative process is repeated across several stages until reaching a training length of $(K_T, K) = (20, 100)$. Empirically, employing multi-stage training also enhances performance for special initialization. With multi-stage training, both initialization approaches yield MiLoDo-trained optimizers with comparable performance, prompting our focus on special initialization due to its reduced warmup stages.

The multi-stage training method draws inspiration from curriculum learning in reinforcement learning, where models are first trained on easier tasks before progressing to more difficult ones. In our context, we initially train the optimizer with a short iteration length, which is easier to train. For instance, consider the extreme case of training an optimizer for just one iteration – this is analogous to training a simple one-layer neural network, which is inherently easier. Once this simpler stage is complete, we gradually extend the iteration length. Starting with easier tasks and then using the results of this stage to initialize the next stage with more complex tasks (longer iteration processes in our context) significantly improves training stability compared to starting directly with difficult tasks from scratch.

### E.2 TARGET PROBLEMS

**LASSO regression.** Decentralized LASSO regression problem with shape $(n, d, N, \lambda)$ is defined as:

$$\min_{\boldsymbol{x} \in \mathbb{R}^d} \quad \frac{1}{2n} \sum_{i=1}^n \|\boldsymbol{A}_i \boldsymbol{x} - \boldsymbol{b}_i\|_2^2 + \lambda \|\boldsymbol{x}\|_1,$$

where $\boldsymbol{A}_i \in \mathbb{R}^{N \times d}$ and $\boldsymbol{b}_i \in \mathbb{R}^N$ are kept on nodes $i$ out of a total of $n$ nodes. To generate LASSO optimizees with shape $(n, d, N, \lambda)$, we first sample $\boldsymbol{A} \in \mathbb{R}^{nN \times d}$ and a vector $\boldsymbol{x}^\star \in \mathbb{R}^d$ from normal distribution. Then, we pick 75% of $\boldsymbol{x}^\star$'s entries with the smallest magnitude and reset them to zero. Afterwards, we generate $\boldsymbol{b} = \boldsymbol{A}\boldsymbol{x}^\star + \epsilon \boldsymbol{z}$, where $\epsilon = 0.1$ is the noise scale and $\boldsymbol{z} \in \mathbb{R}^{nN}$ is sampled from standard Gaussian. Finally, we distribute $\boldsymbol{A}$ and $\boldsymbol{b}$ evenly to each node so that each $\boldsymbol{A}_i \in \mathbb{R}^{N \times d}$ and $\boldsymbol{b}_i \in \mathbb{R}^N$.

**Logistic regression.** Decentralized logistic regression problem with $\ell_1$-regularization and shape $(n, d, N, \lambda)$ is defined as:

$$\min_{\boldsymbol{x} \in \mathbb{R}^d} \quad \frac{1}{n} \sum_{i=1}^n \left[ \frac{1}{N} \sum_{j=1}^N b_{ij} \ln\left(1 + \exp(-\boldsymbol{a}_{ij}^\top \boldsymbol{x})\right) + (1 - b_{ij}) \ln\left(1 + \exp(\boldsymbol{a}_{ij}^\top \boldsymbol{x})\right) \right] + \lambda \|\boldsymbol{x}\|_1,$$

where $\boldsymbol{A}_i = (\boldsymbol{a}_{i1}^\top, \cdots \boldsymbol{a}_{iN}^\top)^\top \in \mathbb{R}^{N \times d}$ and $\boldsymbol{b}_i = (b_{i1}, \cdots, b_{iN})^\top \in \{0, 1\}^N$. To generate synthetic logistic regression optimizees with shape $(n, d, N, \lambda)$, we first sample $\boldsymbol{A} \in \mathbb{R}^{nN \times d}$ and $\boldsymbol{x}^\star \in \mathbb{R}^d$ from normal distribution. Then we pick 75% of $\boldsymbol{x}^\star$'s entries with the smallest magnitude and reset them to zero. Afterwards, we generate $\boldsymbol{b} = (b_1, \cdots, b_{nN})^\top$ by $b_i = \mathbb{1}_{\{\boldsymbol{a}_i^\top \boldsymbol{x}^\star \geq 0\}}$. Finally, we distribute $\boldsymbol{A}$ and $\boldsymbol{b}$ evenly to each node so that each $\boldsymbol{A}_i \in \mathbb{R}^{N \times d}$ and $\boldsymbol{b}_i \in \{0, 1\}^N$.

**MLP training.** We consider a decentralized MLP training problem using MNIST dataset. The model structure is illustrated as in Fig. 12. The total number of trainable parameters in the MLP is 13002. The optimizees are constructed by randomly selecting data from MNIST's training dataset for all nodes.

**ResNet training.** We consider a decentralized ResNet training problem using CIFAR-10 dataset. The model structure is illustrated as in Fig. 13. The total number of trainable parameters in the ResNet model is 78042. The optimizees are constructed by randomly selecting data from CIFAR-10's training dataset for all nodes.

### E.3 IMPLEMENTATION DETAILS

**Model structure.** We use the same model structure throughout our experiments. Specifically, $\phi_{M,i}$ has input dimension 2 and output dimension 1 with ReLU activation, $\phi_{S,i}$ has input dimension $|\mathcal{N}(i)|$ and output dimension $|\mathcal{N}(i)|$ with Exponential activation. $\phi_{U,i}$ has input dimension $|\mathcal{N}(i)|$ and output dimension $|\mathcal{N}(i)|$ with ReLU activation. We use ReLU activation in the middle of the 2-layer MLP. The hidden/output dimensions of the LSTM cells, input/hidden/output dimensions of the MLP are all set to 20.

**Training details.** In our experiments, we employ special initialization and a multi-stage training strategy. As described in Sec. 5, we continually train MiLoDo in five stages with training lengths $(K_T, K) = (5, 10), (10, 20), (20, 40), (40, 80)$ and $(20, 100)$ by Adam with learning rate 5e-04, 1e-04, 5e-05, 1e-05, 1e-05, for 20, 10, 10, 10, 5 epochs, respectively. Throughout all stages, the

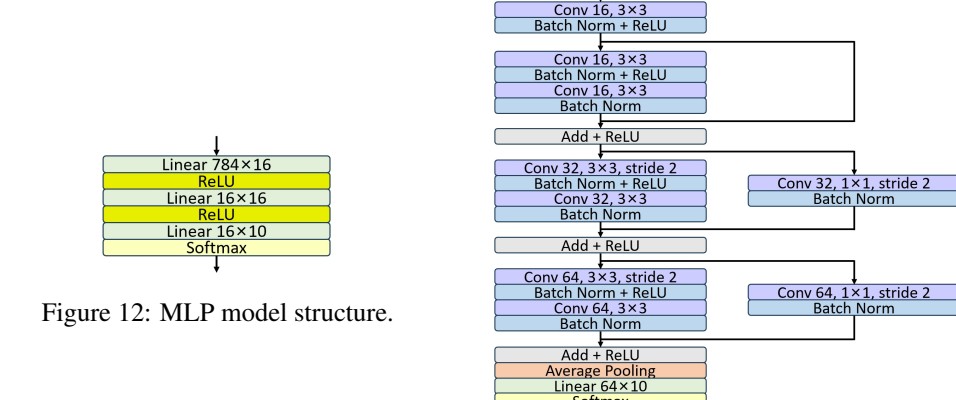

Figure 12: MLP model structure.

Figure 13: ResNet model structure.

Adam optimizer is configured with momentum parameters $(\beta_1, \beta_2) = (0.9, 0.999)$ and the batch size is fixed to 32.

**LASSO with real data.** To generate LASSO$(10, 200, 10, 0.05)$ from BSDS500(Martin et al., 2001) dataset, we first extract a $10 \times 10$ patch from testing images and flatten to vector $\boldsymbol{b} \in \mathbb{R}^{100}$. We normalize $\boldsymbol{b}$ by subtracting the mean. Afterwards, we conduct K-SVD(Aharon et al., 2006) to obtain $\boldsymbol{A} \in \mathbb{R}^{100 \times 200}$. Finally, we distribute $\boldsymbol{A}$ and $\boldsymbol{b}$ evenly to each node so that each $\boldsymbol{A}_i \in \mathbb{R}^{10 \times 200}$ and $\boldsymbol{b}_i \in \mathbb{R}^{10}$. We generate a total of 1000 instances as the testing set in the experiments.

**Construction of meta training set.** As illustrated in Sec. 5, the meta training set consists of synthetic LASSO problems with 20 different shapes: $(10, 500, N, 0.1)$ where $N \in \{5, 10, 15, \cdots, 100\}$. We generate 64 distinct problem instances for each shape, hence 1280 instances in total.

**Evaluation metric.** We evaluate solution $\boldsymbol{X} = [\boldsymbol{x}_1^\top, \boldsymbol{x}_2^\top, \cdots, \boldsymbol{x}_n^\top]^\top$ of decentralized problem (4) via loss $f(\bar{\boldsymbol{x}}) + r(\bar{\boldsymbol{x}})$ and consensus error $\frac{1}{n} \sum_{i=1}^n \|\boldsymbol{x}_i - \bar{\boldsymbol{x}}\|_2$, where $\bar{\boldsymbol{x}} = \frac{1}{n} \sum_{i=1}^n \boldsymbol{x}_i$. All testing curves display averaged performance on 512 instances, except for problems with over 10,000 dimensions which are highly time-consuming to test. For those high-dimensional problems, we display testing performance on a single instance chosen randomly, as results on other instances are quite similar.

**Implementation of baseline algorithms.** Following Appendix D, we have the following robust implementation for the considered baselines, where the learning rate $\gamma$ is manually tuned optimal for each experiment, and we use $W = (w_{ij})_{n \times n}$ with $w_{ij} = 1/3 \cdot \mathbb{1}_{\{i=j, \text{ or } \{i,j\} \in \mathcal{E}\}}$ as the doubly-stochastic goissp matrix for the ring topology.

- **Prox-DGD.** Initialized with $\boldsymbol{x}_i^0 = \boldsymbol{0}_d$, Prox-DGD uses the following update rules:

$$
\boldsymbol{z}_i^{k+1} = \boldsymbol{x}_i^k - \gamma \nabla f_i(\boldsymbol{x}_i^k),
$$

$$
\boldsymbol{x}_i^{k+1} = \text{prox}_{\gamma r}\left(\boldsymbol{z}_i^{k+1} - \sum_{j \in \mathcal{N}(i)} w_{ij}(\boldsymbol{z}_i^{k+1} - \boldsymbol{z}_j^{k+1})\right).
$$

- **Prox-ATC.** Initialized with $\boldsymbol{x}_i^0 = \tilde{\boldsymbol{y}}_i^0 = \boldsymbol{z}_i^0$, Prox-ATC uses the following update rules:

$$
\boldsymbol{z}_i^{k+1} = \boldsymbol{x}_i^k - \gamma \nabla f_i(\boldsymbol{x}_i^k),
$$

$$
\tilde{\boldsymbol{z}}_i^{k+1} = \tilde{\boldsymbol{y}}_i^k - \boldsymbol{z}_i^{k+1} + \boldsymbol{z}_i^k,
$$

$$
\boldsymbol{y}_i^{k+1} = 2\tilde{\boldsymbol{y}}_i^k - \tilde{\boldsymbol{z}}_i^{k+1} + \sum_{j \in \mathcal{N}(i)} w_{ij}(\tilde{\boldsymbol{z}}_i^{k+1} - \tilde{\boldsymbol{z}}_j^{k+1}),
$$

$$
\tilde{\boldsymbol{y}}_i^{k+1} = \boldsymbol{y}_i^{k+1} - \sum_{j \in \mathcal{N}(i)} w_{ij}(\boldsymbol{y}_i^{k+1} - \boldsymbol{y}_j^{k+1}),
$$

$$
\boldsymbol{x}_i^{k+1} = \text{prox}_{\gamma r}(\tilde{\boldsymbol{y}}_i^{k+1}).
$$

- **PG-EXTRA.** Initialized with $\boldsymbol{x}_i^0 = \boldsymbol{0}_d$, PG-EXTRA uses the following update rules:

$$\boldsymbol{z}_i^{k+1} = \boldsymbol{x}_i^k - \sum_{j \in \mathcal{N}(i)} w_{ij}(\boldsymbol{x}_i^k - \boldsymbol{x}_j^k) - \gamma \nabla f_i(\boldsymbol{x}_i^k),$$

$$\tilde{\boldsymbol{z}}_i^{k+1} = \begin{cases} \boldsymbol{z}_i^{k+1}, & \text{if } k = 0, \\ \boldsymbol{z}_i^{k+1} + \tilde{\boldsymbol{z}}_i^k - \boldsymbol{x}_i^{k-1} + \frac{1}{2}\sum_{j \in \mathcal{N}(i)} w_{ij}(\boldsymbol{x}_i^{k-1} - \boldsymbol{x}_j^{k-1}) + \gamma \nabla f_i(\boldsymbol{x}_i^{k-1}), & \text{if } k > 0, \end{cases}$$

$$\boldsymbol{x}_i^{k+1} = \text{prox}_{\gamma r}(\tilde{\boldsymbol{z}}_i^{k+1}).$$

- **Prox-ED.** Initialized with $\boldsymbol{x}_i^0 = \tilde{\boldsymbol{y}}_i^0 = \boldsymbol{z}_i^0 = \boldsymbol{0}_d$, Prox-ED uses the following update rules:

$$\boldsymbol{z}_i^{k+1} = \boldsymbol{x}_i^k - \gamma \nabla f_i(\boldsymbol{x}_i^k),$$

$$\boldsymbol{y}_i^{k+1} = \tilde{\boldsymbol{y}}_i^k + \boldsymbol{z}_i^{k+1} - \boldsymbol{z}_i^k,$$

$$\tilde{\boldsymbol{y}}_i^{k+1} = \boldsymbol{y}_i^{k+1} - \frac{1}{2}\sum_{j \in \mathcal{N}(i)} w_{ij}(\boldsymbol{y}_i^{k+1} - \boldsymbol{y}_j^{k+1}),$$

$$\boldsymbol{x}_i^{k+1} = \text{prox}_{\gamma r}(\tilde{\boldsymbol{y}}_i^{k+1}).$$

**Computational resources.** We conduct all the experiments within a single NVIDIA A100 GPU server with a GPU memory of 80G.

### E.4 ADDITIONAL RESULTS

**Training on logistic regression.** Fig. 14 displays the in-distribution testing results of MiLoDo optimizer trained on a specialized dataset including 512 synthetic Logistic$(10, 50, 100, 0.1)$ optimizees. Fig. 15 displays the testing results of MiLoDo optimizer trained on a specialized dataset including 512 real data Logistic$(10, 14, 100, 0.1)$ optimizees using Census Income (Kohavi, 1996) dataset.

**More testing results of MiLoDo optimizer trained on the meta training set.** As a supplement to the results in Sec. 5, Fig. 16, we further tests MiLoDo trained on the meta training set on synthetic LASSO$(10, 20000, 1000, 0.1)$. While trained on non-smooth optimizees only, the MiLoDo-trained optimizer is consistently fast in solving smooth optimization problems such as linear regression, as illustrated in Fig. 17.

**Testing results of MiLoDo optimizer trained on more complex topologies.** Beyond the findings presented in Sec.5 and Fig.8 (left), further tests were conducted on commonly used topologies. Fig18 demonstrates that MiLoDo optimizer exhibits consistent performance, achieving a 2 to 3 times acceleration, which highlights its scalability and robustness across various topologies.

**Testing results of MiLoDo optimizer trained on a larger network.** Extending the analyses discussed in Sec.5 and illustrated in Fig.8 (right), additional experiments were carried out on networks with 100 nodes. As depicted in Fig.23, MiLoDo optimizer maintained a high level of effectiveness, delivering a $2\times$ to $3\times$ speedup. This not only confirms the optimizer's efficiency but also highlights its scalability and robustness in larger networks.

**Testing results under strict dataset separation strategies.** In order to better validate the generalization ability and performance of MiLoDo optimizer, we further validate its performance on the CIFAR-10 dataset, where data subsets used for training the optimizer, optimizees and computing the test accuracy, are strictly different. As illustrated in Fig.19, MiLoDo optimizer trained on ResNet$(5, 78042, 500, 0)$ performs consistently better than other baseline algorithms on ResNet$(5, 78042, 5000, 0)$.

**Comparison with existing step-size-tuning algorithms.** Existing step-size-tuning algorithms in decentralized optimization, *e.g.*, D-NASA (Li et al., 2024), DADAM (Nazari et al., 2022) and Kuruzov et al., primarily focus on smooth problems. Consequently, we compare MiLoDo optimizer with D-NASA, DADAM and Algorithm 1 in Kuruzov et al. on smooth, LASSO$(10, 300, 10, 0)$ optimizees. As illustrated in Fig.20, MiLoDo optimizer clearly outperforms these baseline algorithms.

**Generalization to higher data heterogeneity.** We tested the training of a 3-layer MLP on MNIST while generating data distributions with varying degrees of heterogeneity using Dirichlet sampling, where the larger the Dirichlet concentration parameter $\alpha$ is, the more identical the distributions are

(Hsu et al., 2019). We trained MiLoDo optimizer on $MLP(10, 13002, 1000, 0)$ using the uniformly distributed MNIST dataset which inherits low data heterogeneity and tested it in high heterogeneity scenarios with $\alpha = 100, 10$ or $1$, to assess its generalization ability. The results in Fig.21 demonstrate that, even without being explicitly trained on highly heterogeneous data, MiLoDo outperforms other algorithms in terms of convergence speed and accuracy. This suggests that MiLoDo does not simply "memorize" the data distribution of specific optimization tasks but instead learns how to adaptively address optimization problems based on their underlying characteristics.

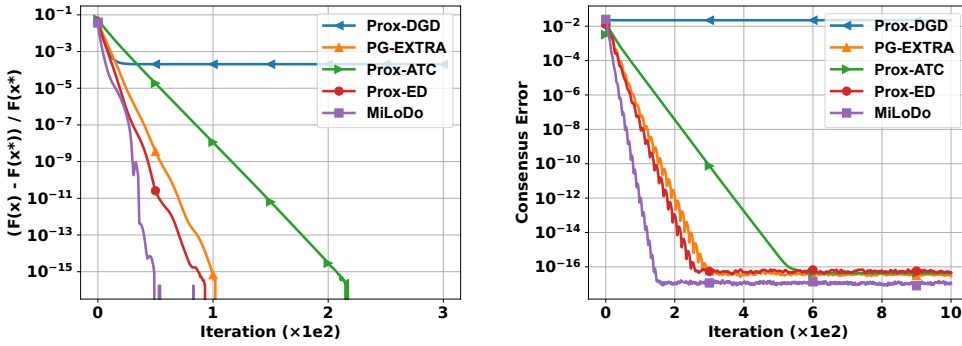

Figure 14: MiLoDo optimizer trained on synthetic $Logistic(10, 50, 100, 0.1)$ and tested on unseen $Logistic(10, 50, 100, 0.1)$ instances.

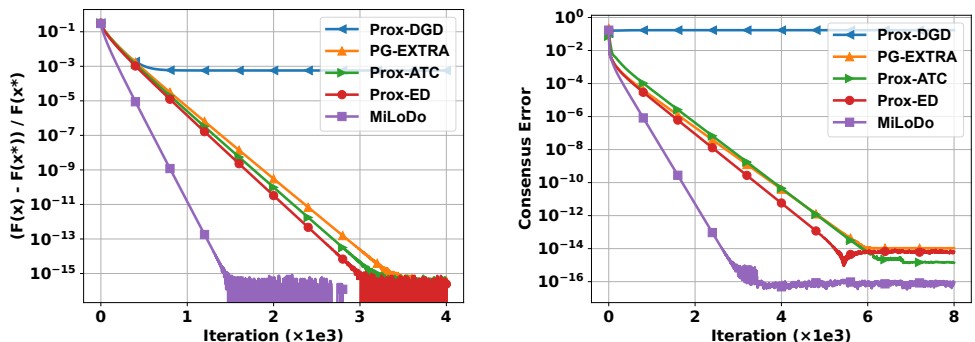

Figure 15: MiLoDo optimizer trained on $Logistic(10, 14, 100, 0.1)$ with Census Income (Kohavi, 1996) dataset and tested on $Logistic(10, 14, 100, 0.1)$ with unseen data in Census Income dataset.

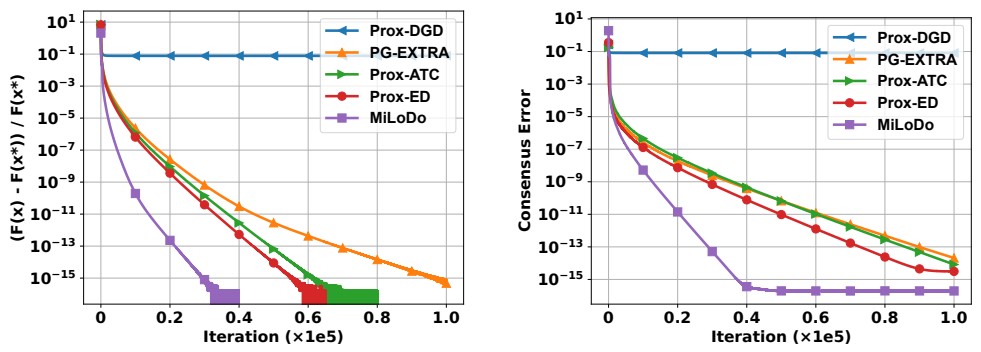

Figure 16: MiLoDo trained on meta learning set and tested on $LASSO(10, 20000, 1000, 0.1)$.

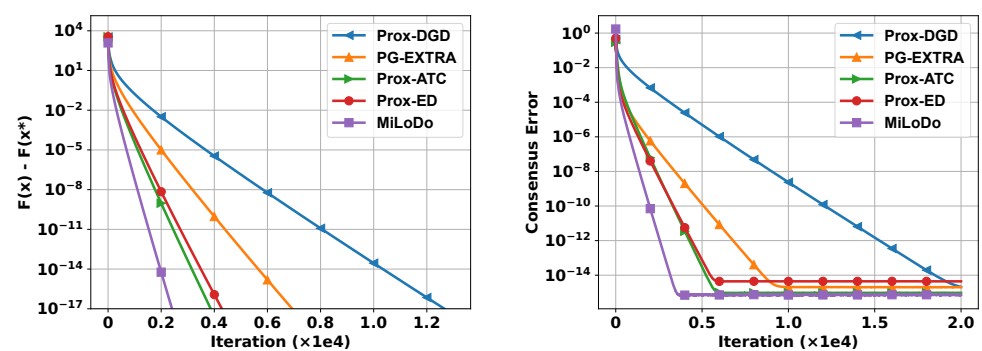

Figure 17: MiLoDo optimizer trained on meta training set and tested on linear regression problems as LASSO(10, 15000, 1000, 0).

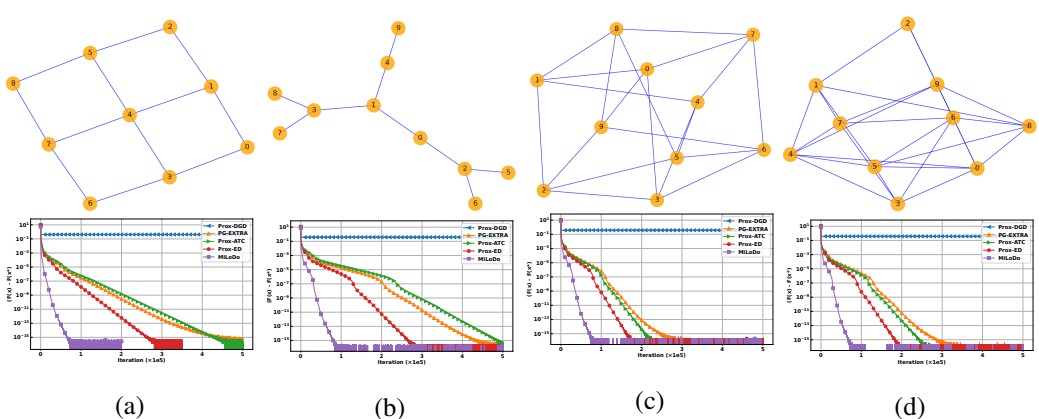

Figure 18: Topology and testing results on (a) LASSO(9,270,10,0.1) on grid topology, (b) LASSO(10,300,10,0.1) on tree topology, (c) LASSO(10,300,10,0.1) on exponential topology, (d) LASSO(10,300,10,0.1) on Erdős-Rényi topology.

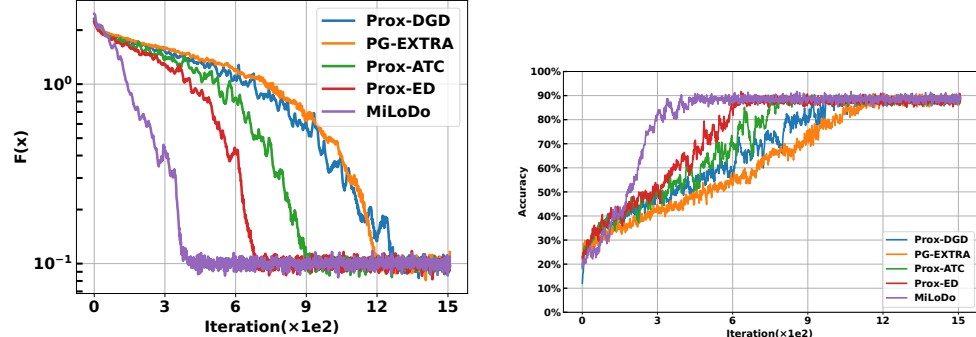

Figure 19: MiLoDo optimizer trained on ResNet(5, 78042, 500, 0) using the CIFAR-10 dataset and evaluated on ResNet(5, 78042, 5000, 0), with strict separation between meta-training and testing datasets.

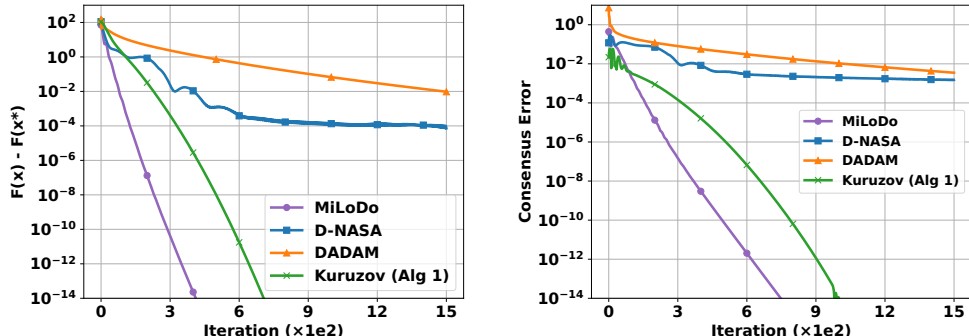

Figure 20: Testing results on LASSO(10,300,10,0.0) of MiLoDo optimizer and step-size-tuning baseline algorithms.

### E.5 ABLATION STUDIES

**Ablation studies on the mixing matrices for baseline algorithms.** The adaptive preconditioners and mixing weights are critical to MiLoDo's performance gain. To better address MiLoDo's advantages, we conduct ablation experiments on the mixing matrices used in the baseline methods. Fig. 22 demonstrates that the performance of using strategically designed and fixed weights (1/3 in our experiments) are almost the same , which provides a stronger validation of MiLoDo 's advantages.

**Ablation on base update rules (12)-(14).** We specify detailed experimental setups for directly learning optimizers from the base update rules (12)-(14), as discussed in Sec. 5. For simplicity, we use $r \equiv 0$ so that the implicit rule in (12) can be explicitly modeled as

$$z_i^{k+1} = x_i^k - m_i^k(\nabla f_i(x_i^k), y_i^k; \theta_{i,1}).$$

Without coordinate-wise structures, the scale of the neural network has to be correlated with the optimizees' dimension. Consequently, we fix the problem dimension $d = 10$ and use LASSO$(10, 10, 5, 0)$ as the training and testing optimizees. We parameterize each of the base update rules with a LSTM model consists of a single LSTM cell and a 2-layer MLP with ReLU activation. The input sizes of the LSTM models are $20$, $10|\mathcal{N}(i)|$, $10|\mathcal{N}(i)|$ for $m_i$, $s_i$, $u_i$, respectively. The output sizes are 10 according to the problem dimension. All hidden dimensions in the LSTM cells and MLPs are set to 100. We use random initialization and multi-stage training strategy similar to MiLoDo to train the parameterized base update rules.

**Ablation studies on the multi-stage training method.** The stable training of optimizers in the Learning to Optimize field is a widely recognized challenge. A commonly used approach is the single-stage training strategy, where the optimizer is trained for many epochs with a fixed training length of $(K_T, K) = (20, 100)$. However, we observed that this approach is highly sensitive to the choice of training hyperparameters. Specifically, when the learning rate is too small or the number of

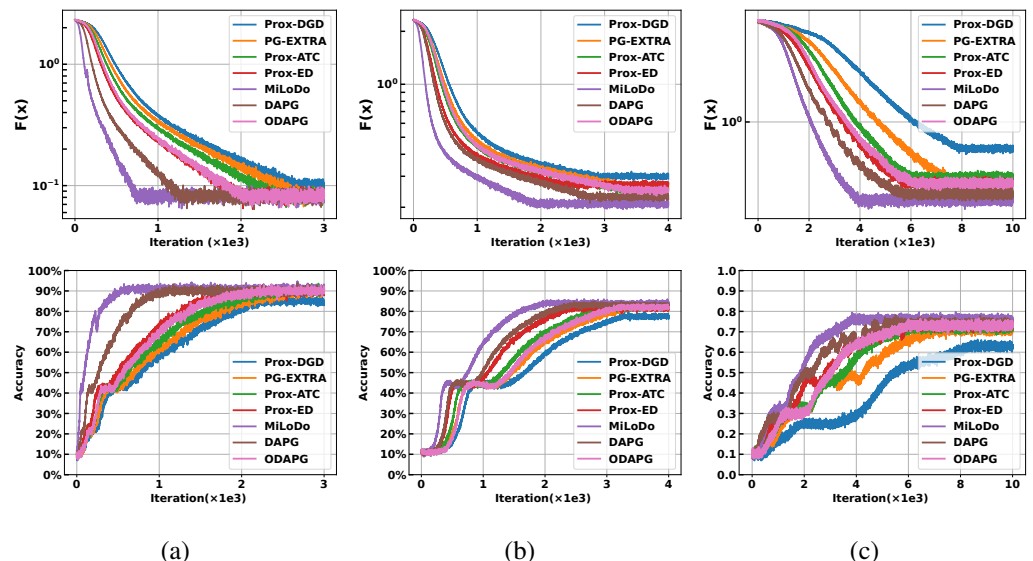

(a)          (b)          (c)

Figure 21: MiLoDo optimizer trained on MLP(10, 13002, 1000, 0) using the MNIST dataset and tested on MLP(10, 13002, 3000, 0) under varying heterogeneity levels: (a) $\alpha = 100$, (b) $\alpha = 10$, (c) $\alpha = 1$.

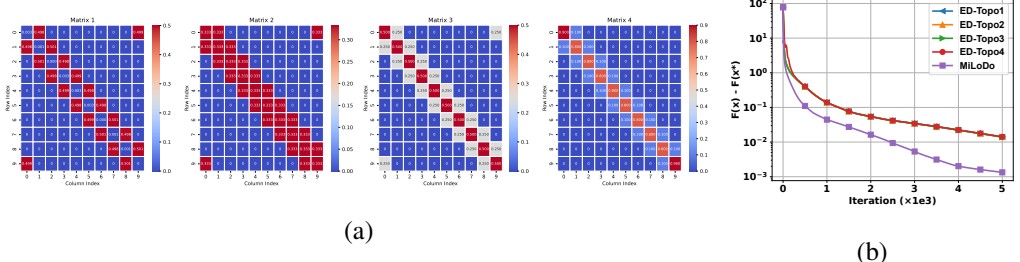

(a)

(b)

Figure 22: **Ablation on mixing matrices.** (a) describes different choices of mixing matrices, where matrix 1 is computed by solving the FMMC problem via projected subgradient algorithm (Boyd et al., 2004); (b) displays testing results of solving LASSO problem by Prox-ED with different mixing matrices, showcasing that the 1/3-strategy (matrix 2) is already good enough.

epochs is insufficient, the model tends to underfit. Conversely, when the learning rate or the number of epochs is too large, the loss may explode during training, leading to instability. This hyperparameter sensitivity poses significant challenges to the stable and reliable training of the MiLoDo optimizer.

To address these challenges, we propose a multi-stage training strategy designed to improve training stability by dividing the training process into multiple stages with distinct objectives. To validate the effectiveness of this approach, we conducted ablation studies comparing the conventional single-stage training method and the proposed multi-stage training strategy. Specifically, we evaluated the performance of trained optimizers on the LASSO optimization problem $(10, 300, 100, 0.1)$ under various hyperparameter settings, with detailed configurations summarized in Table 2.

In the single-stage training strategy, where $(K_T, K) = (20, 100)$ is fixed, we observed that small changes in the learning rate or the number of epochs resulted in significant variations in performance, as shown in Fig 24a. This highlights the high sensitivity of this approach to hyperparameter configurations. In contrast, the multi-stage training strategy demonstrated significantly reduced sensitivity by dividing the process into multiple stages, as shown in Fig 24b. For simplicity, we ablated only the hyperparameters of the first stage. This simplification is reasonable, as our observations indicate that early-stage training has a critical impact on the final performance. Thus, modifying hyperparameters in the first stage alone is sufficient to validate the robustness and stability of the multi-stage training strategy without compromising the reliability of our conclusions.

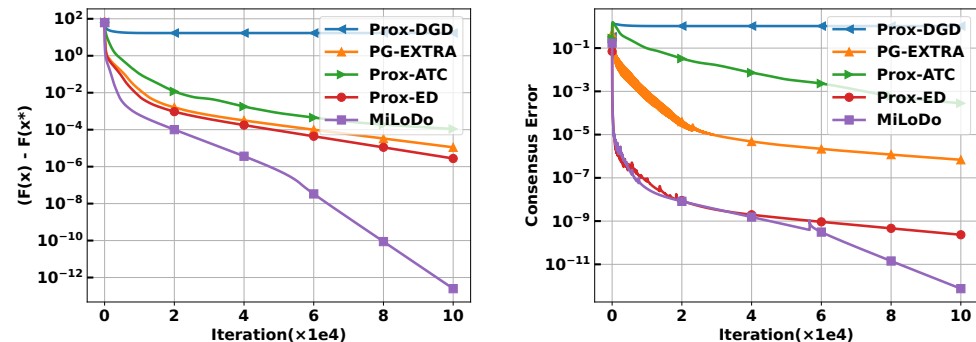

Figure 23: MiLoDo optimizer trained on a large network with 100 nodes, and tested on LASSO(10, 300, 10, 0.1).

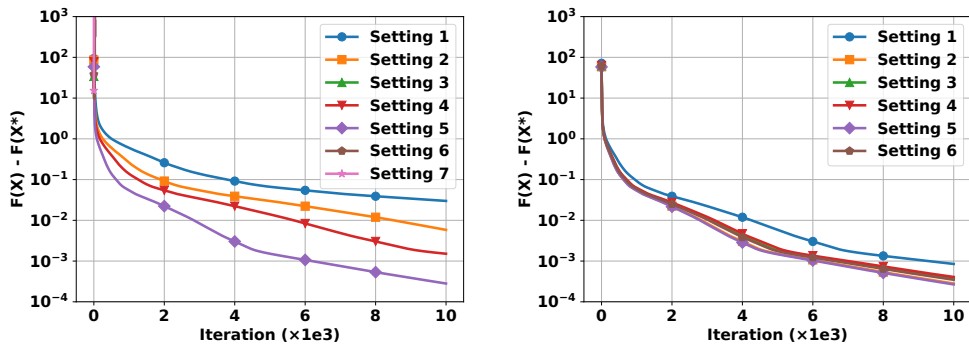

(a) Single-stage training: the optimizer is trained with a fixed training length of $(K_T, K) = (20, 100)$ while varying the learning rate and number of epochs.

(b) Multi-stage training: the optimizer is trained in multiple stages, and ablation studies are conducted by modifying the hyperparameters of only the first stage.

Figure 24: Comparison of hyperparameter settings between single-stage and multi-stage training.

Table 2: Hyperparameter settings in single-stage and multi-stage training strategies.

| Single-Stage Training ($K_T, K = 20, 100$) | | |
| --- | --- | --- |
| **Setting** | **Epochs** | **Learning Rate** |
| 1 | 20 | 0.0001 |
| 2 | 40 | 0.0001 |
| 3 | 60 | 0.0001 |
| 4 | 20 | 0.0005 |
| 5 | 40 | 0.0005 |
| 6 | 60 | 0.0005 |
| 7 | 20 | 0.001 |
| Multi-Stage Training ($K_T, K = 5, 10$) | | |
| **Setting** | **Epochs** | **Learning Rate** |
| 1 | 20 | 0.0001 |
| 2 | 20 | 0.0005 |
| 3 | 20 | 0.001 |
| 4 | 40 | 0.0001 |
| 5 | 40 | 0.0005 |
| 6 | 40 | 0.001 |

## E.6 HYPERPARAMETER SETTINGS

We specify the manually-tuned optimal learning rates of baseline algorithms for all the experiments in Table 3.

Table 3: Optimal learning rates of baseline algorithms chosen in different experiments.

| Experiment | Prox-ED | PG-EXTRA | Prox-ATC | Prox-DGD | DAPG | ODAPG |
|---|---|---|---|---|---|---|
| LASSO$(10, 300, 10, 0.1)$ | 0.03 | 0.02 | 0.025 | 0.04 | 0.01 | 0.02 |
| LASSO$(10, 30000, 1000, 0.1)$ | 0.03 | 0.02 | 0.025 | 0.04 | 0.01 | 0.02 |
| LASSO$(10, 200, 10, 0.1)$ | 0.05 | 0.04 | 0.045 | 0.05 | 0.02 | 0.03 |
| LASSO$(10, 20000, 1000, 0.1)$ | 0.05 | 0.04 | 0.045 | 0.05 | / | / |
| LASSO$(10, 15000, 1000, 0.0)$ | 0.08 | 0.05 | 0.085 | 0.09 | / | / |
| Logistic $(10, 50, 100, 0.1)$ | 1.0 | 0.8 | 0.4 | 1.0 | / | / |
| Logistic $(10, 14, 100, 0.1)$ | 1.9 | 1.7 | 1.8 | 2.0 | / | / |
| MLP$(10, 13002, 5000, 0)$ | 0.09 | 0.06 | 0.06 | 0.05 | 0.03 | 0.055 |
| ResNet$(5, 78042, 5000, 0)$ | 0.1 | 0.07 | 0.08 | 0.05 | 0.05 | 0.07 |

