# OpenReview forum: "A Mathematics-Inspired Learning-to-Optimize Framework for Decentralized Optimization"
_ICLR.cc/2025/Conference — Submitted to ICLR 2025_

### Official Review · Reviewer_ebwG · 2024-10-19

**Soundness:** 2
**Presentation:** 2
**Contribution:** 2
**Rating:** 3
**Confidence:** 4

**Summary:**

This paper proposes an approach to extend the Learning to Optimize framework to the decentralized setting. The authors compared the trained model with many commonly used decentralized optimization approaches using empirical evaluations.

**Strengths:**

The effort of extending Learning to Optimize framework to the decentralized setting is interesting.

**Weaknesses:**

1. The comparison with existing decentralized optimization results are not fair. It is commonly known that existing decentralized optimization algorithms are sensitive to parameters such as learning rate. How did you select learning rate for these existing decentralized optimization algorithms? If these existing algorithms were to given the best parameters, then their performance cannot be beaten by the proposed algorithm, which uses essentially the same algorithmic structure.

2. It is well known that exiting decentralized algorithms are sensitive to parameters like learning rate, it seems much easier to learn the learning rate, than to conduct the learning proposed here.

**Questions:**

see weakness.

---

> ### Author Response · Authors · 2024-11-22
> **Response to Reviewer ebwG**
>
> Thank you for reviewing our work and for providing feedback. We would like to offer the following clarifications in response to your concerns:
>
> **[W1] How to Choose the Learning Rate**: We acknowledge that existing decentralized optimization algorithms are highly sensitive to the choice of learning rate. Therefore, we made substantial efforts on tuning these learning rates to make sure the comparison is fair. Specifically, we randomly generated 256 optimization problems (serving as a "training set") for each type of optimization problem and conducted global learning rate tuning to identify the optimal learning rate. We believe this method ensures that the chosen learning rate performs well across a wide range of optimization problems, and thus we are confident that our comparison is fair. (Refer to Appnedix E.6 for details.)
>
> **[W1] MiLoDo is clearly different from merely tuning learning rates**: Our method, MiLoDo (Equations 22, 23, 24), differs significantly from classic approaches in three key aspects: **(I)** $\mathbf{p}\_i^k$ is not just a learning rate, but a diagonal matrix; **(II)** In addition to $\mathbf{p}\_i^k$, we tune (or learn) the gossip weights $\mathbf{p}\_{i,j,1}^k,\mathbf{p}\_{i,j,2}^k$; **(III)** All tunable components are parameterized by neural networks, allowing them to adapt dynamically to the iterative process and optimization problem, without relying on predefined parameters. Therefore, we respectfully disagree with the reviewer's point "If these existing algorithms were to given the best parameters, then their performance cannot be beaten by the proposed algorithm, which uses essentially the same algorithmic structure."
>
> **[W2] Why Not Simply Tune the Learning Rates?** While we agree with the reviewer that merely tuning the learning rate is easy to implement, our proposed method offers two key advantages:
> - As discussed in Points I and II above, our method incorporates additional parameters and components to tune (or learn), increasing the algorithm's capacity and improving the potential for better performance. Our experiments in Section 5 demonstrate that this potential can be effectively realized with the proposed training method.
> - As described in Point III, all tunable components are automatically adjusted based on the current state of the iterative process, making the overall algorithm less sensitive to initial choices. This adaptivity enhances robustness, as fixed parameters may perform well for one instance but fail for others.
>
> We hope these clarifications address your concerns. If you have further questions or require additional clarifications, we would be more than happy to provide them. We greatly appreciate it if you could update your evaluation should you find these responses satisfactory.

---

> > ### Comment · Reviewer_ebwG · 2024-11-22
> >
> > I read the response and retain my opinion.

---

> ### Author Response · Authors · 2024-11-25
>
> Dear Reviewer ebwG,
>
> We greatly appreciate your feedback and would like to clarify a few points to ensure our responses are fully understood.
>
> **1. Comparison with baseline algorithms and how MiLoDo achieves better results**
>
> As noted in Appendix E.6, we manually tuned the learning rates for all baseline algorithms to ensure optimal performance in every experiment.
>
> MiLoDo consistently outperforms these baselines due to the following reasons:
>
> (a) MiLoDo is a more general algorithmic framework with vectorized learning rates, providing stronger representational capabilities.
>
> (b) Unlike traditional methods, MiLoDo adapts learning rates and gossip matrices dynamically, treating them as part of the optimization process rather than fixed hyperparameters.
>
> The consistent performance improvements demonstrated in our experiments further validate MiLoDo’s effectiveness.
>
> **2. Why not simply tune learning rates?**
>
> Figure 20 shows that MiLoDo significantly outperforms existing step-size-tuning algorithms. This highlights that MiLoDo’s adaptivity and generality offer advantages that cannot be replicated by simple learning rate tuning.
>
> **3. MiLoDo optimizer eliminates the need for hyperparameter tuning, reducing effort and time consumption**
>
> Traditional handcrafted algorithms often require extensive hyperparameter tuning to achieve competitive performance, as what we have done in our experiments. This process can be both effort-intensive and time-consuming.
>
> In contrast, MiLoDo eliminates the need for such manual tuning. Once trained, it can be applied directly without setting any hyperparameters, as it automatically adjusts all step sizes and gossip matrices during the optimization process. This makes MiLoDo more efficient and user-friendly compared to traditional approaches.
>
> In our response, we have clearly articulated why learning-to-optimize can outperform hand-tuning, supported by strong numerical evidence demonstrating the strengths of our framework. However, we find the rationale behind the reviewer's decision unclear. The reviewer’s perspective seems highly subjective and lacks substantiating evidence. We would greatly appreciate it if the reviewer could outline specific concerns or questions regarding our rebuttal and experiments, enabling us to address them more effectively.

---

### Official Review · Reviewer_XKvB · 2024-10-30

**Soundness:** 3
**Presentation:** 3
**Contribution:** 2
**Rating:** 6
**Confidence:** 4

**Summary:**

This paper addresses the learning to optimize problem in the decentralized setting. The current learning to optimize algorithms suffer from big search space and poor generalization. The paper considers the mathematical conditions requires in the decentralized system to achieve consensus and optimality and proposed MiLoDo. For a fixed network with individually trained agents, MiLoDo demonstrated robust performance with generalization properties for both synthetic and real data.

**Strengths:**

The MiLoDo framework provides insights on the necessary mathematical conditions for ensuring consensus and optimality in decentralized training, which could be of separate interest.

This work bridges the L2O literature with decentralized optimization techniques, the problem formulation is intuitive and the simplification on parameters as well as reductions in parameter search space works well in practice.

I have not checked the technical details of the proof in the paper's appendix, but the analytical results in the main paper seem to be intuitive and aligns with the existing literature in the decentralized optimization literature.

**Weaknesses:**

Perhaps one of the most important aspects of decentralized optimization and training over a network, as noted in papers such as EXTRA and various gradient tracking approaches, is to address the data and/or function heterogeneity across agents. In the numerical experiments, especially for MNIST and CIFAR-10, the data seems to be evenly split across agents. When the loss functions are similar across the decentralized network, the system essentially reduces to a SVRG problem, where the consensus and optimality questions are, in a sense, trivial. This paper would benefit from additional experiments regarding the impact of data heterogeneity in the framework. The actual notion of generalization should also be discussed regarding whether MiLoDo can generalize across various degrees of heterogeneity. I will base my opinion on the acceptance of this paper on how the authors address this question in their rebuttal.

As the authors have mentioned, the current MiLoDo framework requires a fixed network with fully synchronized updates. This assumption is difficult to satisfy for many current applications of decentralized optimization tasks.

MiLoDo requires a set of parameters to be learned for all connections among agents, which suggests that the worst-case complexity of the problem is O(n^2) with respect to network size. This scaling is undesirable for large-scale applications.

Despite the mathematical inspiration behind MiLoDo, the framework did not directly address the impact of network structure e.g. the connectivity of the decentralized network, with the only assumption that the graph is strongly connected.

Currently each agent in the MiLoDo framework is required to learn individual parameters, which are specific to the agents. A more concise and elegant solution would be training one set of parameters which can be used for all agents in the network. Though this might not satisfy the mathematical conditions noted in the papers, some discussions would be appreciated.

**Questions:**

Please see the weaknesses.

---

> ### Author Response · Authors · 2024-11-22
> **Response to Reviewer XKvB (Part 1/2)**
>
> Thank you for your detailed review and valuable feedback. Your comments have been immensely helpful in improving our paper and strengthening the experimental design. Below, we provide point-by-point responses to your concerns and suggestions:
>
>
> 1. **On the Issue of Data Heterogeneity**:
> We greatly appreciate your attention to the issue of data and function heterogeneity. Indeed, due to the incomplete data assigned to each node, our current experiments inherently include some level of heterogeneity. However, we acknowledge that the degree of heterogeneity may not fully reflect the complexity of real-world scenarios. To address this, we have conducted additional experiments to evaluate the performance of MiLoDo on heterogeneous data. Specifically, we tested the training of a 3-layer MLP on MNIST while generating data distributions with varying degrees of heterogeneity using Dirichlet sampling. In these experiments, **MiLoDo was trained under low heterogeneity settings and tested in high heterogeneity scenarios** to assess its generalization ability.
> **The results and discussions are highlighted in Appendix E.4 of our revised paper, titled "Generalization to higher data heterogeneity."** The results demonstrate that, even without being explicitly trained on highly heterogeneous data, MiLoDo outperforms other algorithms in terms of convergence speed and accuracy. This suggests that MiLoDo does not simply "memorize" the data distribution of specific optimization tasks but instead learns how to adaptively address optimization problems based on their underlying characteristics.
>
>
> 2. **On the Assumptions of Fixed Network and Synchronous Updates**:
> We understand your concerns regarding the assumptions of a fixed network topology and synchronous updates. These assumptions may indeed be challenging to meet in certain distributed optimization tasks. We would like to clarify that the current setup of fixed networks and synchronous updates was chosen to ensure a fair comparison with existing distributed optimization algorithms, such as EXTRA, which operate under similar assumptions. Nevertheless, we acknowledge that these assumptions have limitations in practical applications. In future work, we plan to extend MiLoDo to handle scenarios with dynamic network topologies and asynchronous updates, thereby broadening its applicability to more realistic and complex distributed settings.
>
>
>
> 3. **On Time Complexity and Scalability to Network Size**:
> We acknowledge that for complete graphs, the single-iteration complexity would indeed be $\mathcal{O}(n^2)$ where $n$ is the number of nodes in the graph. However, for sparse graphs, the complexity is proportional to the number of edges: $\mathcal{O}(|\mathcal{E}|)$, as all operations are either local or involve only neighboring nodes. For instance, in Equation (22), all computations are performed locally; in Equations (23) and (24), each node communicates solely with its neighbors, where $\mathcal{N}(i)$ denotes the set of neighbors for node $i$. Thus, the overall complexity is $\mathcal{O}(|\mathcal{E}|)$ rather than $\mathcal{O}(n^2)$. Additionally, in practical scenarios, large-scale graphs are more likely to be sparse, as it is uncommon for every pair of nodes in a large graph to be directly connected.
>
>
> 4. **On the Impact of Network Structure**:
> First, we would like to clarify that a strongly-connected graph is different from a complete graph. In a complete graph, every pair of nodes is directly connected by an edge, whereas in a strongly-connected graph, we only require that every pair of nodes is connected through a path. This is a much weaker assumption. In our paper, we assume a strongly-connected network to ensure the feasibility and convergence of our algorithm.
> While we did not directly analyze the impact of network topology on optimization performance, MiLoDo has shown the ability to learn effective communication strategies and update rules under various topologies. Regarding the "impact of network structure," we understand that aspects such as sparsity, degree distribution, and other topological properties may have a significant influence on optimization performance. *In the appendix (Section E.4, "Testing results of MiLoDo optimizer trained on more complex topologies."),* we have included **experimental results under various network structures, which show that MiLoDo exhibits strong adaptability** and learning capability, consistently outperforming hand-designed algorithms across different settings.
> We would appreciate it if you could further explain "the framework did not directly address the impact of network structure" and we would like to provide further clarification on this point.

---

> > ### Comment · Reviewer_XKvB · 2024-12-02
> >
> > I have read the response and am glad my feedback was useful. However, there are some questions I feel not sufficiently addressed, even with the rebuttal.
> >
> > The original experiments did not address the heterogeneity problem, while the authors provided additional experiments, using a model trained on homogeneous dataset on heterogeneous task still did not address the triviality problem as mentioned previously.
> >
> > I will be keeping my score as is.

---

> ### Author Response · Authors · 2024-11-22
> **Response to Reviewer XKvB (Part 2/2)**
>
> 5. **On Parameter Sharing**:
> Currently, MiLoDo learns separate parameters for each neighbor relationship (i.e., communication links), allowing communication and update rules to be optimized specifically for each connection. We fully agree with the reviewer that "A more concise and elegant solution would be training one set of parameters which can be used for all agents in the network." We appreciate this suggestion regarding parameter sharing and recognize its potential as an improvement direction. In future work, we will investigate the feasibility of parameter sharing and aim to develop a more compact framework that combines shared and personalized parameters to further enhance the efficiency and generalization of MiLoDo.
> In fact, if the same set of parameters were shared across all nodes and edges, the overall algorithm described in Equations (22)-(24) and (25)-(27) would resemble a **message-passing graph neural network (GNN)**. GNNs are not only more scalable but also offer additional advantages, such as permutation-equivariance, which makes them particularly well-suited for graph-based problems. We have incorporated this discussion into the revised paper. (Section 6)
>
>
>
> We sincerely thank the reviewer for the thoughtful and constructive comments. We hope these responses address your concerns and clarify the contributions of our work. We look forward to further discussions and would be more than happy to address any additional questions or feedback.

---

### Official Review · Reviewer_TVRE · 2024-11-03

**Soundness:** 3
**Presentation:** 3
**Contribution:** 3
**Rating:** 6
**Confidence:** 4

**Summary:**

This paper introduces a new approach to learning-to-optimize for distributed optimization. The core contribution is a new parameterization of the optimizer update rule which is motivated by theoretical conditions that any update rule must satisfy for convergence. This is an elegant approach leading to a new method, MiLoDo, that exhibits superior empirical convergence behavior compared to prior work in the literature on small test problems (LASSO, logistic regression, MLP/MNIST, and ResNet/CIFAR).

**Strengths:**

1. The paper is well-written and easy to follow.
2. The contributions are well-situated with respect to the recent literatures on learned optimizers and decentralized optimization.
3. The motivation for the parameterization introduced in equations (12)--(14) as well as the implementation in (22)--(24) is sound and interesting.
4. Thorough experiments with small problems illustrate the promise of this approach.
5. That MiLoDo generalizes to many more iterations than the learned optimizer was trained on is impressive.

**Weaknesses:**

1. The motivation to use decentralized optimizers remains somewhat unclear, and there doesn't appear to be a well-known widely-adopted implementation today. This may change in the future, especially as some companies talk about training models across data centers. However, without that, the overall motivation for this contribution remains limited.
2. The experimental results focus on smaller problems, the largest being a ResNet trained on CIFAR. These are all problems that can be easily solved today in a centralized manner with very modest/accessible compute resources. The empirical results would be much stronger if they demonstrated that the same trends hold at scales where distributed/decentralized training is necessary.
3. Some experimental details are unclear; for example is the same 5-stage MiLoDo training procedure used for all workloads, including MLP and ResNet? How sensitive is MiLoDo training to this procedure? The MiLoDo training setup for MNIST and CIFAR is also less convincing, given that the meta-training is done using (subsets of) the same dataset that evaluation is performed on. It would be much more convincing to see results on neural network training where there is clear separation between training ("optimizees") and the workload used to evaluate the learned optimizer.

**Questions:**

1. For Lines 82--93, are there references that could be cited to support the last three challenges mentioned, especially "weak generalization"?
2. Is there stronger motivation that can be provided for where decentralized (learned) optimization is being applied or will be useful in practice? Similarly, can motivation be provided for decentralized LASSO or logistic regression tasks?
3. Could consistent notation be used throughout the paper? In Sections 2 and 3 the learnable update rule is denoted $\bf{g}$, while in Section 4 $\bf{g}$ is used for subgradients of the regularization term and the learned update rule is $\bf{m}$.
4. I'm confused how the generalization to problems with higher problem dimension works. From Section 4.2 I understand that the update rule is implemented as an LSTM, for which I would have thought the shapes of its weights depended on the problem dimension. Can you please explain?
5. The formulation and derivation of MiLoDo in Section 4.1 is written with $p_i^k$, $p_{i,j,1}^k$ and $p_{i,j,2}^k$ that are memoryless, while in Section 4.2 we see these are implemented with LSTMs, i.e., RNNs that have memory/state across iterations. That this helps or is useful is not surprising since it is known that optimizers with memory/momentum are important for efficient convergence of decentralized optimization (see, e.g., EXTRA, DIGing, SlowMo). How do we reconcile this important difference between Sections 4.1 and 4.2? Is memory helpful/sufficient, but not necessary according to the theory? Do the theoretical claims also apply to this implementation with LSTMs?
6. Any intuition about the loss spike/instability (in $F(x)$) observed in Fig 7 around iteration 500?
7. The experiments illustrate the effectiveness and capabilities of trained MiLoDo optimizers. It would be good to also include some illustration and discussion about the process of training MiLoDo optimizers to provide more intuition and support for the five-stage procedure.
8. I appreciated the inclusion of runtimes and transparency about MiLoDo and learned optimizers requiring more computation per update. Please include more information about the system on which timing experiments were run for results in Table 1. It is impossible to interpret these times without knowing more about the system setup.

---

> ### Author Response · Authors · 2024-11-22
> **Response to Reviewer TVRE (Part 1/3)**
>
> Thank you for your detailed review of our work and for your valuable feedback. Your comments have been immensely helpful in improving our paper. Below, we provide a point-by-point response to your concerns and suggestions:
>
> **[W1] On the Motivation for Decentralized Optimization:** We understand your concerns regarding the motivation for decentralized optimization. Existing research shows that decentralized optimization offers significant advantages in scenarios where data is distributed, communication bandwidth is limited, or data privacy is critical. Additionally, decentralized optimization reduces dependency on central nodes, alleviating computational resource bottlenecks, which is one of the key motivations behind our work.
>
> We have included the following references and discussions in the revised paper (Section 1):
> > Decentralized optimization has become a standard paradigm for distributed training without centralizing data (Liu et al., 2024), and its significant advantages in communication efficiency (Lian et al., 2017) and privacy protection (Yu et al., 2024) have made it potential in privacy-preserving distributed learning across data centers.
>
> **[W2] On the Scale of Experiments and the Motivation for Distributed Setup:** We acknowledge your concern about the scale of the experiments. Although training ResNet on CIFAR can be easily done on a single GPU, we chose this experiment to use a standardized evaluation environment, allowing for fair comparisons with existing optimization algorithms and to validate the effectiveness of our algorithm on typical tasks. Due to the constraints of our lab environment, conducting large-scale tests is challenging. However, this does not hurt the primary contribution of our work: introducing a learning-based decentralized optimization framework with mathematical guarantees.
>
> **[W3] On the Separation of Evaluation Datasets:** Thank you for your suggestion regarding dataset separation. We fully agree that the evaluation dataset should be separate from the meta-training dataset. To address this, we have added a new experiment in which we train ResNet on CIFAR with strict adherence to dataset separation, ensuring that different subsets are used for meta-training, model training, and evaluation.
>
> *The results and discussions are presented in Appendix E.4 of our revised paper, titled "Testing results under strict dataset separation strategies."* MiLoDo demonstrates superior performance in this more challenging setting, further validating its generalization ability and the effectiveness of the MiLoDo optimizer.
>
> **[W4] On the Applicability and Sensitivity of MiLoDo’s Five-Stage Training:** The same five-stage training process is a general framework applicable to **all optimization tasks presented in our paper**, including MLP and ResNet. Actually, compared with a a vanilla end-to-end training, this multi-stage training method is more stable and less sensitive, particularly to initialization. This is the primary reason we adopted this approach. In contrast, vanilla end-to-end training sometimes requires sophisticated initialization (e.g., initializing the optimizer to mimic a traditional one), whereas the five-stage training entirely eliminates this need. The same procedure can be consistently applied across various tasks.
>
> We understand the reviewer's concern: a sophisticated algorithm might be sensitive. However, the five-stage training framework avoids this issue. To clarify, the method draws inspiration from curriculum learning in reinforcement learning, where **models are first trained on easier tasks before progressing to more difficult ones**. In our context, we initially train the optimizer with a short iteration length, which is easier to train. For instance, consider the extreme case of training an optimizer for just one iteration—this is analogous to training a simple one-layer neural network, which is inherently easier. Once this simpler stage is complete, we gradually extend the iteration length. Starting with easier tasks and then using the results of this stage to initialize the next stage with more complex tasks (longer iteration processes in our context) significantly improves training stability compared to starting directly with difficult tasks from scratch.
>
> We have included the above discussions in the updated draft. (Appendix E.1)

---

> > ### Comment · Reviewer_TVRE · 2024-11-26
> >
> > Thanks for your responses. I'm glad to hear my feedback was useful.
> >
> > While I appreciate the responses, none has convinced me sufficiently that any of the concerns were addressed to the point where I would change my score, hence I'm keeping my score as is.
> >
> > In particular,
> >
> > *[W1]* The motivations mentioned in the response, while true, are too high-level and not specifically addressing my comment. Moreover, the connection of this work to private optimization or federated learning is not clear at all given that privacy preserving mechanisms like secure aggregation or differential privacy are not incorporated in the algorithm description, experiments, or analysis.
> >
> > *[W4]* To convincingly argue that the five-stage procedure makes training more robust, it would be useful to include experiments where various aspects of the setup are perturbed to illustrate the effects and how they do/don't impact performance.

---

> > > ### Author Response · Authors · 2024-12-01
> > >
> > > We sincerely thank you for your thoughtful review and valuable feedback. We deeply appreciate the time and effort you have taken to evaluate our work and provide constructive comments. Below, we address your concerns in detail:
> > >
> > >
> > > **[W1] Motivation for Decentralized Optimization**
> > >
> > > In centralized networks (e.g., parameter-server architectures), all nodes must communicate with a central node for synchronization. This incurs a total communication cost of **O(n)**, where **n** is the number of nodes. In contrast, decentralized networks rely on peer-to-peer local interactions, where each node communicates only with its neighbors. The communication cost in this setup is determined by the maximum degree of the graph (**d**), resulting in a total communication cost of **O(d)**. For sparse graph topologies (e.g., ring or grid structures), **d** becomes a constant, and the communication cost can be reduced to **O(1)**.
> > >
> > > This reduction in communication cost is one of the key motivations for decentralized optimization, as it alleviates communication bottlenecks that often arise in large-scale systems. For example, in the work of Lian et al. (2017), the authors analyzed the Decentralized Parallel Stochastic Gradient Descent (D-PSGD) algorithm and demonstrated that it achieves the same convergence rate (or equivalently, computational complexity) as the Centralized Parallel Stochastic Gradient Descent (C-PSGD) algorithm. However, D-PSGD significantly outperforms C-PSGD's communication efficiency by avoiding the "communication traffic jam" caused by transmitting data to a central node. This result demonstrates the unique advantages of decentralized optimization in distributed learning systems. Empirically, Figure 2 in Lian et al. (2017) shows that the decentralized algorithm has a $3\times$ speedup compared with two centralized implementations in training wallclock time on a 7-GPU network.
> > >
> > > One of the key motivations for our work stems from the communication benefits, and our algorithm design leverages the decentralized topology to achieve efficient optimization. Furthermore, in many real-world distributed computing scenarios, network topologies are inherently sparse (e.g., sensor networks, peer-to-peer networks), which further amplifies the advantages of decentralized optimization. We hope this more specific explanation addresses your concerns regarding the motivation for decentralized optimization.
> > >
> > > **[W4] Robustness of the Five-Stage Training Procedure**
> > >
> > > Yes, we do observe that the five-stage training procedure is more robust to hyperparameters in training. Detailed results are presented in our revised paper, **_Appendix E.5_**, under the paragraph "Ablation studies on the multi-stage training method," and illustrated in Figure 24.
> > >
> > > To summarize briefly: with a standard single-stage training procedure, as shown in **Fig. 24a**, small changes in the learning rate or the number of epochs resulted in significant performance variations, highlighting the instability of this method. In contrast, the proposed five-stage training strategy significantly reduced this sensitivity. As shown in **Fig. 24b**, the five-stage procedure consistently achieved strong performance as long as the hyperparameters were within a reasonable range. This finding demonstrates that the five-stage training procedure not only stabilizes the training process but also lowers the complexity of hyperparameter tuning.
> > >
> > > We believe these results provide strong evidence that the proposed five-stage procedure improves training robustness and addresses the hyperparameter sensitivity issue commonly encountered in single-stage training. We hope this additional explanation and the updated results in the paper address your concern.
> > >
> > >
> > > Finally, we would like to thank you again for your constructive feedback, which has helped us improve the clarity and rigor of our work. Should you have any further concerns, we would be happy to address them.

---

> ### Author Response · Authors · 2024-11-22
> **Response to Reviewer TVRE (Part 2/3)**
>
> **[Q1] On Literature Supporting the Challenges:** Yes, the "weak generalization" is supported by the survey and benchmark paper (Chen et al. "Learning to optimize: A primer and a benchmark." JMLR 2022.), which has been cited in our paper. Specifically, Section 4.4 and Figures 7 and 8 in that paper provide support for the points made in our introduction. We have revised the paragraph to clarify this reference.
>
> **[Q2] On the Motivation for Decentralized LASSO/Logistic:** In scenarios like hospital collaborations, predictive models can be trained without exposing sensitive patient data by sharing partial information, such as gradients. However, this approach is not entirely secure, as patient data can still be inferred from the shared gradients. To address this, many decentralized learning frameworks, such as federated learning, aggregate gradients from multiple nodes before sharing or using them. This aggregation helps obscure individual contributions, reducing the risk of inferring specific data points.
>
> **[Q3] On Notion Consistency:** Thank you for pointing out the issue of inconsistent symbols. In the revised version, we have addressed this by ensuring consistent notation, using the letter $m$ to denote update rules and $g$ to represent subgradients.
>
>
> **[Q4] On Generalization to Problems with Higher Dimension:** In our framework, the shape of the LSTM weights is independent of the problem's dimensionality because we utilize a **coordinate-wise LSTM** approach. Specifically, for each coordinate in the variables $\mathbf{x},\mathbf{y},\mathbf{z}$, a separate LSTM is applied, and these LSTMs share common weights across coordinates. When inputting data into the LSTM, the problem's dimensionality is mapped to the `batch_size` dimension, which does not affect the shape of the LSTM weights.
>
> Now, a natural question might arise: with so many LSTMs, what is the computational overhead? In practice, the computational cost of LSTMs is typically much smaller than that of gradient computations. We apply a separate LSTM for each coordinate, so the complexity is $\mathcal{O}(n)$. For example, if $f(\mathbf{x})=\\|\mathbf{A}\mathbf{x}-\mathbf{b}\\|^2$ with $\mathbf{A}\in\mathbb{R}^{m \times n}$, computing the gradient $\mathbf{A}^\top(\mathbf{A}\mathbf{x}-\mathbf{b})$ involves a complexity of $\mathcal{O}(mn)$. This comparison is supported experimentally by Table 1 on Page 10.
>
> **[Q5] LSTM’s Incorporation of Historical Information:** While Section 4.2 incorporates historical information, and it might seem unsurprising that an optimizer leveraging such information performs better, it could appear that this approach increases the "power" of the optimizer. However, we argue that the reality is quite the opposite: from Section 4.1 to Section 4.2, the "power" of the optimizer is actually reduced!
>
> Consider the following simplified example:
> $$\min_{p_1,p_2} f(x_1)+f(x_2) ~~~~ \textup{ s.t. } x_1 = x_0 - p_1 \nabla f(x_0), ~~ x_2 = x_1 - p_2 \nabla f(x_1) $$
> and
> $$
> \begin{align}
> \min_{\theta} f(x_1)+f(x_2) ~~~~ \textup{ s.t. } & x_1 = x_0 - p_1 \nabla f(x_0), ~~ x_2 = x_1 - p_2 \nabla f(x_1) \\\\
> & p_1,h_1 = \phi(x_0,h_0; \theta), ~~ p_2,h_2 = \phi(x_1,h_1; \theta)
> \end{align}
> $$
> In the first problem, we directly optimize over the values of $p_0,p_1$, ensuring that the objective value is theoretically optimal (or at least equivalent, given a sufficiently large neural network $\phi$). In the second formulation, $p_0,p_1$ are obtained indirectly via parameterization through $\phi(\cdot;\theta)$, which may lead to suboptimal results.
>
> This example illustrates the relationship between Sections 4.1 and 4.2. Section 4.1 introduces a theoretical optimizer that is not practical because it requires storing all $\mathbf{p}_{i}^k$ matrices. Memorizing these matrices is unscalable as the number of iterations $k$ increases. In contrast, Section 4.2 presents a practical parameterization approach. While the parameterized optimizer may not achieve the same theoretical optimum as the approach in Section 4.1, it is more scalable. Furthermore, our experiments demonstrate that this parameterized optimizer is not only practical but also very effective.
>
>
> **[Q6] On the Loss Spike in ResNet Training:** The loss spike observed in Figure 7 during ResNet training may be related to dynamically changing input information. Similar to traditional optimizers that use dynamic learning rates, a brief spike in loss may occur, followed by a reduction. MiLoDo quickly adjusts through its learned update rules, ultimately achieving global convergence.

---

> ### Author Response · Authors · 2024-11-22
> **Response to Reviewer TVRE (Part 3/3)**
>
> **[Q7] Intuition behind Five-Stage Training:** Please see our response in "[W4] On the Applicability and Sensitivity of MiLoDo’s Five-Stage Training"
>
> **[Q8] On System Information for Time Measurement:** In our experiments, we used a single NVIDIA A100 80GB GPU, simulating multiple nodes on a single card. We measured the time each node took for one iteration, averaged it across all nodes, and then accumulated this to obtain the total time for multiple iterations.The detailed device specifications are as follows:
>
> - CPU: Dual Intel® Xeon® Silver 4310 processors with 48 logical cores (2.10 GHz base frequency, 3.30 GHz turbo boost) and a total of 36 MiB L3 cache distributed across two NUMA nodes.
> - Memory: 125 GiB DDR4 RAM, with 109 GiB available during the experiments.
> - GPU: NVIDIA A100 80GB PCIe GPU (Driver Version: 535.183.01, CUDA Version: 12.2) featuring 80 GB of high-bandwidth HBM2e memory, utilized for all computations.
> - Operating System: Ubuntu 22.04.4 LTS.
>
> Once again, thank you for your thorough review and valuable suggestions. We will incorporate your feedback in the revised version to further improve the paper’s content and experimental design.

---

### Official Review · Reviewer_r2et · 2024-11-04

**Soundness:** 2
**Presentation:** 2
**Contribution:** 2
**Rating:** 3
**Confidence:** 3

**Summary:**

This paper proposes distributed algorithm called MiLoDo that solve consensus-type problems. Some simulations are run to demonstrate the speed of convergence.

**Strengths:**

- Distributed optimization is an important problem.
- This paper is fairly easy to read and the methods seem to make sense.
- A number of simulations are done.

**Weaknesses:**

- It seems that not much in the paper is new? A lot of the tricks are standard in distributed optimization, for example, variable duplicating to create equality constraints are used in ADMM.
- I'm also not sure what mathematical-inspired means. The methods are follow standard approaches.
- I would suggest that the paper do a better job in describing what is different between the material in the paper and existing work out there.

**Questions:**

- The update rules in this paper is similar to tuning the step size in iterative algorithms. There are a lot of different methods for tuning these step sizes, it would be useful to compare against some of these.

---

> ### Author Response · Authors · 2024-11-22
> **Response to Reviewer r2et (Part 1/2)**
>
> We thank the reviewer for the comments and have made every effort to address the concerns raised. Below, we provide a detailed response and clarifications.
>
> - **(Novelty).** The reviewer expressed concerns about the novelty of the paper, suggesting similarities to step-size tuning. However, we believe the contribution of this paper is far beyond merely tuning step sizes and standard methods (to the best of our knowledge). We explain our motivation and contributions in detail below:
> - **(Motivation).** Since the reviewer mentioned step-size tuning, we’ll start by discussing it and then expand to broader concepts. While tuning step sizes might seem straightforward, it is actually non-trivial to design an adaptive policy that adjusts step sizes based on the features of the iterative process. For instance, what are the necessary and sufficient conditions for such a policy to ensure convergence? In other words, what defines a good step-size tuning strategy?
> Beyond merely tuning step sizes, can we replace the step size with an adaptive preconditioner matrix to improve performance? Even more ambitiously, can we adaptively tune the gossip matrix (communication strategies between nodes on a graph) based on the iterative process's features? Unfortunately, to our knowledge, these questions are not fully addressed in the context of decentralized optimization.
> - **(Contributions regarding "Mathematics-inspired").** Given the wide range of components that can be tuned in practice, as discussed above, we directly assume that the **entire algorithm can be tuned** (or learned) from data. Specifically, we consider the following general scheme (equations (12-14) in our paper):
> $$\begin{align}
> \mathbf{z}\_i^{k+1}=\ &\mathbf{x}\_i^k-\mathbf{m}\_i^k(\nabla f\_i(\mathbf{x}\_i^k),\mathbf{g}\_i^{k+1},\mathbf{y}\_i^k),\quad\mathbf{g}\_i^{k+1}\in\partial r(\mathbf{z}\_i^{k+1}),\\\\
> \mathbf{y}\_i^{k+1}=\ &\mathbf{y}\_i^k+\mathbf{s}\_i^k(\\{\mathbf{z}\_i^{k+1}-\mathbf{z}\_j^{k+1}\\}\_{j\in\mathcal{N}(i)}),\\\\
> \mathbf{x}\_i^{k+1}=\ &\mathbf{z}\_i^{k+1}-\mathbf{u}\_i^k(\\{\mathbf{z}\_i^{k+1}-\mathbf{z}\_j^{k+1}\\}\_\{j\in\mathcal{N}(i)\}).
> \end{align}$$
> where $\mathbf{m}^k_i(\cdot)$, $\mathbf{s}^k_i(\cdot)$ and $\mathbf{u}^k_i(\cdot)$ are general mappings without particular structures and they will be learned from data! This approach is inspired by the paradigm of Learning to Optimize (L2O), which differs significantly from traditional decentralized optimization methods.
> Now A natural question arises: **what conditions should these mappings ($\mathbf{m}^k_i$, $\mathbf{s}^k_i$ and $\mathbf{u}^k_i$) satisfy to ensure convergence?** This is a key contribution of our work. In Theorem 1, we show that convergence requires specific structures for these mappings, formalized in Equations (17)-(19). Furthermore, in Theorem 2, we show that, all the fixed points of iterative algorithm described by (17)-(19) must be optimal solutions.
> Therefore, our findings provide a foundational principle when we want to learn an decentralized optimizer from data: if we require convergence of the algorithm, mappings $\mathbf{m}^k_i$, $\mathbf{s}^k_i$ and $\mathbf{u}^k_i$ must satisfy minimal yet essential conditions derived from mathematical analysis (Conditions 1 and 2, Theorems 1 and 2). This is why we describe Equations (17)-(19) as "mathematics-inspired," in contrast to the purely data-driven approach represented by Equations (12)-(14).
> - **(Differences from classic methods).** Besides the above contributions, our proposed method MiLoDo (22,23,24) is clearly different from classic methods in three aspects: (I) $\mathbf{p}\_i^k$ is not just a step size, but a diagonal matrix; (II) In addition to $\mathbf{p}\_i^k$, we tune (or learn) the gossip weights $\mathbf{p}\_{i,j,1}^k,\mathbf{p}\_{i,j,2}^k$; (III) All tunable components are parameterized by neural networks, allowing them to adapt dynamically to the iterative process and optimization problem, without relying on predefined parameters. Finally, experimental results demonstrate the superior performance of MiLoDo.
> - **(Conclusion).** Based on the discussion above, we believe that our findings and contributions are both novel and significant. While we draw inspiration from standard approaches (Lines 216-241), the concepts and methods introduced from Section 3.2 onward are new.

---

> ### Author Response · Authors · 2024-11-22
> **Response to Reviewer r2et (Part 2/2)**
>
> **(Added experiments: Comparison with step size tuning).** To further address the reviewer's concern, we have added more comparison experiments, particularly against state-of-the-art algorithms for adaptive step-size tuning. **The results and discussions are detailed in Appendix E.4 of our revised paper, titled "Comparison with Existing Step-Size-Tuning Algorithms."** As shown in Fig. 20, the MiLoDo optimizer consistently outperforms these algorithms.
>
> We hope these clarifications address your concerns. If you have further questions or require additional clarifications, we would be more than happy to provide them. We greatly appreciate it if you could update your evaluation should you find these responses satisfactory.

---

### Meta-Review · Area_Chair_cBG5 · 2024-12-12

**Metareview:**

This paper considers a learning to optimize (L2O) task for the distributed optimization problems. However, a straight-forward extension of the L2O framework to distributed optimization has difficulties in both the huge size of the search space as well as the lack of a mechanism to ensure consensus. Therefore, instead of searching among the general form first-order algorithm space, the authors propose to limit the space of primal-dual first-order algorithm space. To further simplify the search space, the authors propose to learn how to do diagonal scaling (coordinate-style learning rates tuning like adam) for primal-dual method.  Based on the basic form of primal-dual algorithm, the authors design a learn-to-optimize scheme and train the MiLoDo optimizer. Numerical result has shown the advantage of proposed MiLoDo meta-optimizer.

However, this work also has some drawbacks.

1. The work is a relatively simple extension from L2O to the distributed setting. The only methodology novelty is properly restricting the search space to some relatively well-known algorithmic space. E.g. primal-dual first-order algorithms with (bounded and positive) diagonal scaling.

2. This work only inherits the methodology from L2O but has no attempt to resolve the drawbacks of L2O. For example, the L2O meta-optimizer is trying to overfitting the training dynamics of the training problem, while having many limitations in generalizing to different problems. E.g., the optimizer learned to trained ResNet may not work well for CNN and RNN. In fact, as the proposed framework does not consider any dimension-agnostic representations (such as coordinate-wise algorithmic framework), the trained optimizer may not even work for the same problem with different problem dimension. The authors' experiment is a bit cheating in this point as their meta-dataset for LASSO is 20000 dimensional and one may pad the redundant dimensions with zeros in the input, then they can work on different dimension such as 50, 100, etc. But this is definitely not the correct approach. The authors provide additional experiment that samples from the same MNIST dataset with different sampling distribution, but this does not seem enough, and reviewers are not convinced by such experiments. A good meta-optimizer should have the ability to generalize across different datasets. If the meta-optimizer learned on MNIST does not work as well on CIFAR, we cannot say it has good generalization w.r.t. data heterogeneity. Moreover, as the distributed feature of the algorithm introduces additional network topology to the algorithm, this also cause generalization issue due to the change of network topology.   Any small change in the problem setting may require additional retraining of the meta-learner.  All these issues limit the application of the method to practical needs.

Overall, we think the work is marginally below the acceptance bar and we decide to reject the paper.

**Additional Comments On Reviewer Discussion:**

There are several major issues raised by reviewers.

1. Novelty of MiLoDo as a simple extension of L2O.

Though the authors have provided explanation on this. The AC does not think the novelty issue is well-justified. See Meta-Review. Though the authors justify that their work is not a simple learning rate tuning, their Theorem 1 + (21)-(24) indicate that the meta-optimizer is essentially trying to do some adaptive diagonal scaling, which is tuning learning rates in a coordinate style.

2. A range of experimental issues.

   2.1. Lack of experimental details.        Well-resolved by adding detailed explanation for experiments and dataset settings.

   2.2. Need more data heterogeneity.      Partially resolved by adding new experiment on the training of a 3-layer MLP on MNIST while generating data distributions with varying degrees of heterogeneity using Dirichlet sampling. However, this is essentially training the same MNIST dataset, which is well-clustered and too simple. Both the reviewer and the AC is not fully convinced by this.

The other issues are mostly minor and have been addressed by the authors.

---

### Decision · Program_Chairs · 2025-01-22

Reject